# Beating SGD Saturation with Tail-Averaging and Minibatching

**Nicole Mücke**
Institute for Stochastics and Applications
University of Stuttgart
nicole.muecke@mathematik.uni-stuttgart.de

**Gergely Neu**
Universitat Pompeu Fabra
gergely.neu@gmail.com

**Lorenzo Rosasco**
Universita' degli Studi di Genova
Istituto Italiano di Tecnologia
Massachusetts Institute of Technology
lrosasco@mit.edu

## Abstract

While stochastic gradient descent (SGD) is one of the major workhorses in machine learning, the learning properties of many practically used variants are still poorly understood. In this paper, we consider least squares learning in a nonparametric setting and contribute to filling this gap by focusing on the effect and interplay of multiple passes, mini-batching and averaging, in particular tail averaging. Our results show how these different variants of SGD can be combined to achieve optimal learning rates, providing practical insights. A novel key result is that tail averaging allows faster convergence rates than uniform averaging in the nonparametric setting. Further, we show that a combination of tail-averaging and minibatching allows more aggressive step-size choices than using any one of said components.

## 1 Introduction

Stochastic gradient descent (SGD) provides a simple and yet stunningly efficient way to solve a broad range of machine learning problems. Our starting observation is that, while a number of variants including multiple passes over the data, mini-batching and averaging are commonly used, their combination and learning properties are studied only partially. The literature on convergence properties of SGD is vast, but usually only one pass over the data is considered, see, e.g., [23]. In the context of nonparametric statistical learning, which we consider here, the study of one-pass SGD was probably first considered in [35] and then further developed in a number of papers (e.g., [37, 36, 25]). Another line of work derives statistical learning results for one pass SGD with averaging from a worst-case sequential prediction analysis [29, 18, 28]. The idea of using averaging also has a long history going back to at least the works of [32] and [27], see also [34] and references therein. More recently, averaging was shown to lead to larger, possibly constant, step-sizes, see [2, 10, 11]. A different take on the role of (weighted) averaging was given in [24], highlighting a connection with ridge regression, a.k.a. Tikhonov regularization. A different flavor of averaging called *tail averaging* for one-pass SGD was considered in [19] in a parametric setting. The role of minibatching has also being considered and shown to potentially lead to linear parallelization speedups, see e.g. [7] and references therein. Very few results consider the role of multiple passes for learning. Indeed, this variant of SGD is typically analyzed for the minimization of the empirical risk, rather than the actual population risk, see for example [4]. To the best of our knowledge the first paper to analyze the learning properties of multipass SGD was [31], where a cyclic selection strategy was

considered. Other results for multipass SGD were then given in [16] and [20]. Our starting point are the results in [21] where optimal results for multipass SGD where derived considering also the effect of mini-batching. Following the approach in this latter paper, multipass SGD with averaging was analyzed by [26] with no minibatching.

In this paper, we develop and improve the above results on two fronts. On the one hand, we consider for the first time the role of multiple passes, mini-batching and averaging at once. On the other hand, we further study the beneficial effect of tail averaging. Both mini-batching and averaging are known to allow larger step-sizes. Our results show that their combination allows even more aggressive parameter choices. At the same time averaging was shown to lead to slower convergence rates in some cases. In a parametric setting, averaging prevents linear convergence rates [2, 11]. In a nonparametric setting, it prevents exploiting the possible regularity in the solution [10], a phenomenon called *saturation* [12]. In other words, uniform averaging can prevent optimal rates in a nonparametric setting. Our results provide a simple explanation to this effect, showing it has a purely deterministic nature. Further, we show that tail averaging allows to bypass this problem. These results parallel the findings of [19], showing similar beneficial effects of tail-averaging and minibatching in the finite-dimensional setting. Following [21], our analysis relies on the study of batch gradient descent and then of the discrepancy between batch gradient and SGD, with the additional twist that it also considers the role of tail-averaging. The rest of the paper is organized as follows. In Section 2, we describe the least-squares learning problem that we consider, as well as the different SGD variants we analyze. In Section 3, we collect a number of observations shedding light on the role of uniform and tail averaging. In Section 4, we present and discuss our main results. In Section 5 we illustrate our results via some numerical simulations. Proofs and technical results are deferred to the appendices.

## 2   Least Squares Learning with SGD

In this section, we introduce the problem of supervised learning with the least squares loss and then present SGD and its variants.

### 2.1   Least squares learning

We let $(X, Y)$ be a pair of random variables with values in $\mathcal{H} \times \mathbb{R}$, with $\mathcal{H}$ a real separable Hilbert space. This latter setting is known to be equivalent to nonparametric learning with kernels [31]. We focus on this setting since considering infinite dimensios allows to highlight more clearly the regularization role played by different parameters. Indeed, unlike in finite dimensions, regularization is needed to derive learning rates in this case. Throughout the paper we will suppose that the following assumption holds:

**Assumption 1.** *Assume $\|X\| \leq \kappa$, $|Y| \leq M$ almost surely, for some $\kappa, M > 0$.*

The problem of interest is to solve

$$\min_{w \in \mathcal{H}} \mathcal{L}(w), \qquad \mathcal{L}(w) = \frac{1}{2} \mathbb{E}[(Y - \langle w, X \rangle)^2] \tag{1}$$

provided a realization $x_1, \ldots, x_n$ of $n$ identical copies $X_1, \ldots, X_n$ of $X$. Defining

$$\Sigma = \mathbb{E}[X \otimes X], \quad \text{and} \quad h = \mathbb{E}[XY], \tag{2}$$

the optimality condition of problem (1) shows that a solution $w_*$ satisfies the normal equation

$$\Sigma w_* = h. \tag{3}$$

Finally, recall that the excess risk associated with any $w \in \mathcal{H}$ can be written as [1]

$$\mathcal{L}(w) - \mathcal{L}(w_*) = \left\| \Sigma^{1/2}(w - w_*) \right\|^2.$$

## 2.2 Learning with stochastic gradients

We now introduce various gradient iterations relevant in the following. The basic stochastic gradient iteration is given by the recursion

$$w_{t+1} = w_t - \gamma_t x_t(\langle x_t, w_t \rangle - y_t)$$

for all $t = 0, 1 \ldots$, with $w_0 = 0$. For all $w \in \mathcal{H}$ and $t = 1, \ldots n$,

$$\mathbb{E}[X_t(\langle X_t, w \rangle - Y_t)] = \nabla L(w), \tag{4}$$

hence the name. While the above iteration is not ensured to decrease the objective at each step, the above procedure and its variants are commonly called Stochastic Gradient Descent (SGD). We will also use this terminology. The sequence $(\gamma_t)_t > 0$, is called step-size or learning rate. In its basic form, the above iteration prescribes to use each data point only once. This is the classical stochastic approximation perspective pioneered by [30].

In practice, however, a number of different variants are considered. In particular, often times, data points are visited multiple times, in which case we can write the recursion as

$$w_{t+1} = w_t - \gamma_t x_{i_t}(\langle x_{i_t}, w_t \rangle - y_{i_t}).$$

Here $i_t = i(t)$ denotes a map specifying a strategy with which data are selected at each iteration. Popular choices include: *cyclic*, where an order over $[n]$ is fixed a priori and data points are visited multiple times according to it; *reshuffling*, where the order of the data points is permuted after all of them have been sampled once, amounting to sampling without replacement; and finally the most common approach, which is sampling each point *with replacement* uniformly at random. This latter choice is also the one we consider in this paper. We broadly refer to this variant of SGD as *multipass-SGD*, referring to the "multiple passes" 'over the data set as $t$ grows larger than $n$.

Another variant of SGD is based on considering more than one data point at each iteration, a procedure called *mini-batching*. Given $b \in [n]$ the mini-batch SGD recursion is given by

$$w_{t+1} = w_t + \gamma_t \frac{1}{b} \sum_{i=b(t-1)+1}^{bt} (\langle w_t, x_{j_i} \rangle - y_{j_i}) x_{j_i} \,,$$

where $j_1, ..., j_{bT}$ are i.i.d. random variables, distributed according to the uniform distribution on $[n]$. Here the number of *passes* over the data after $t$ iterations is $\lceil bt/n \rceil$. Mini-batching can be useful for at least two different reasons. The most important is that considering mini-batches is natural to make the best use of memory resources, in particular when distributed computations are available. Another advantage is that in this case more accurate gradient estimates are clearly available at each step.

Finally, one last idea is considering averaging of the iterates, rather than working with the final iterate,

$$\bar{w}_T = \frac{1}{T} \sum_{t=1}^{T} w_t.$$

This is a classical idea in optimization, where it is known to provide improved convergence results [32, 27, 15, 2], but it is also used when recovering stochastic results from worst case sequential prediction analysis [33, 17]. More recently, averaging was shown to lead to larger step-sizes, see [2, 10, 11]. In the following, we consider a variant of the above idea, namely *tail-avaraging*, where for $0 \leq S \leq T - 1$ we let

$$\bar{w}_{S,T} = \frac{1}{T - S} \sum_{t=S+1}^{T} w_t \,.$$

We will occasionally write $\bar{w}_L = \bar{w}_{S,T}$, with $L = T - S$. In the following, we study how the above ideas can be combined to solve problem (1) and how such combinations affect the learning properties of the obtained solutions.

## 3 An appetizer: Averaging and Gradient Descent Convergence

Averaging is known to allow larger step-sizes for SGD but also to slower convergence rates in certain settings [10]. In this section, we present calculations shedding light on these effects. In particular,

we show how the slower convergence is a completely deterministic effect and how *tail* averaging can provide a remedy. In the rest of the paper, we will build on these reasonings to derive novel quantitative results in terms of learning bounds. The starting observation is that since SGD is based on stochastic estimates of the expected risk gradient (cf. equations (1), (4)) it is natural to start from the exact gradient descent to understand the role played by averaging.

For $\gamma > 0$, $w_0 = 0$, consider the population gradient descent iteration,

$$u_t = u_{t-1} - \gamma \mathbb{E}[X(\langle X, u_{t-1} \rangle - Y)] = (I - \gamma \Sigma) u_{t-1} + \gamma h,$$

where the last equality follows from (2). Then using the normal equation (3) and a simple induction argument [12], it is easy to see that,

$$u_T = g_T(\Sigma) \Sigma w_*, \qquad g_T(\Sigma) = \gamma \sum_{j=0}^{T-1} (I - \gamma \Sigma)^j. \qquad (5)$$

Here, $g_T$ is a *spectral filtering* function corresponding to a truncated matrix geometric series (the von Neumann series). For the latter to converge, we need $\gamma$ such that $\|I - \gamma \Sigma\| < 1$, e.g. $\gamma < 1/\sigma_M < 1/\kappa^2$, with $\sigma_M = \sigma_{max}(\Sigma) \le \kappa^2$, hence recovering a classical step-size choice. The above computation provides a way to analyze gradient descent convergence. Indeed, one can easily show that

$$w_* - u_T = r_T(\Sigma) w_*, \qquad r_T(\Sigma) = (I - \gamma \Sigma)^T$$

since $g_T(\Sigma)\Sigma = (I - (I - \gamma\Sigma)^T)w_*$, from basic properties of the Neumann series defining $g_T$.

The properties of the so-called *residual operators* $r_T(\Sigma)$ control the convergence of GD. Indeed, if $\sigma_m = \sigma_{min}(\Sigma) > 0$, then

$$\left\| \Sigma^{1/2}(u_T - w_*) \right\|^2 = \left\| \Sigma^{1/2} r_T(\Sigma) w_* \right\|^2 \le \sigma_M (1 - \gamma \sigma_m)^{2T} \|w_*\| \le \sigma_M e^{-2\sigma_m \gamma T} \|w_*\|^2,$$

from the basic inequality $1 + z \le e^z$, highlighting that the population GD iteration converges exponentially fast to the risk minimizer. However, a major caveat is that assuming $\sigma_{min}(\Sigma) > 0$ is clearly restrictive in an infinite dimensional (nonparametric) setting, since it effectively implies that $\Sigma$ has finite rank. In general, $\Sigma$ will not be finite rank, but rather compact with 0 as the only accumulation point of its spectrum. In this case, it is easy to see that the slower rate

$$\left\| \Sigma^{1/2}(u_T - w_*) \right\|^2 \le \frac{1}{\gamma T} \|w_*\|^2$$

holds without any further assumption on the spectrum, since one can show, using spectral calculus and a direct computation [2], that $s^{1/2} r_T(s) \le 1/\gamma T$. It is reasonable to ask whether it is possible to interpolate between the above-described slow and fast rates by making some intermediate assumption. Raher than making assumption on the spectrum of $\Sigma$, one can assume the optimal solution $w_*$ to belong to a subspace of the range of $\Sigma$, more precisely that

$$w_* = \Sigma^r v_* \qquad (6)$$

holds for some $r \ge 0$ and $v_* \in \mathcal{H}$, where larger values of $r$ correspond to making more stringent assumptions. In particular, as $r$ goes to infinity we are essentially assuming $w_*$ to belong to a finite dimensional space. Assumption (6) is common in the literature of inverse problems [12] and statistical learning [8, 9]. Interestingly, it is also related to so-called conditioning and Łojasiewicz conditions, known to lead to improved rates in continuous optimization, see [13] and references therein. Under assumption (6), and using again spectral calculus, it is possible to show that, for all $r \ge 0$,

$$\left\| \Sigma^{1/2}(u_T - w_*) \right\|^2 = \left\| \Sigma^{1/2} r_T(\Sigma) \Sigma^r v_* \right\|^2 \lesssim \left( \frac{1}{\gamma T} \right)^{2r+1} \|v_*\|^2.$$

Thus, higher values of $r$ result in faster convergence rates, at the price of more stringent assumptions.

## 3.1 Tail averaged gradient descent

Given the above discussion, we can derive analogous computations for (tail) averaged GD and draw some insights. Using (5), for $S < T$, we can write the tail-averaged gradient

$$\bar{u}_{S,T} = \frac{1}{T-S} \sum_{t=S+1}^{T} u_t \tag{7}$$

as

$$\bar{u}_{S,T} = G_{S,T}(\Sigma)\Sigma w_*, \qquad G_{S,T}(\Sigma) = \frac{1}{T-S} \sum_{j=S+1}^{T} g_t(\Sigma). \tag{8}$$

As before, we can analyze convergence considering a suitable residual operator

$$w_* - u_{S,T} = R_{S,T}(\Sigma)w_*, \qquad R_{S,T}(\Sigma) = I - G_{S,T}(\Sigma)\Sigma \tag{9}$$

which, in this case, can be shown to take the form,

$$R_{S,T}(\Sigma) = \frac{(I-\gamma\Sigma)^{S+1}}{\gamma(T-S)} (I - (I-\gamma\Sigma)^{T-S})\Sigma^{-1}$$

and where with an abuse of notation we denote by $\Sigma^{-1}$ the pseudoinverse of $\Sigma$. The case of uniform averaging corresponds to $S = 0$, in which case the residual operator simplifies to

$$R_{0,T}(\Sigma) = \frac{(I-\gamma\Sigma)}{\gamma T} (I - (I-\gamma\Sigma)^{T})\Sigma^{-1}.$$

When $\sigma_m > 0$, the residual operators behave roughly as

$$\|R_{S,T}(\Sigma)\|^2 \approx \frac{e^{-\sigma_m \gamma(S+1)}}{\gamma(T-S)}, \qquad \|R_{0,T}(\Sigma)\|^2 \approx \frac{1}{\gamma T},$$

respectively. This leads to a slower convergence rate for uniform averaging and shows instead how tail averaging with $S \propto T$ can preserve the fast convergence of GD.

When $\sigma_m = 0$, taking again $S \propto T$, it is easy to see by spectral calculus that the residual operators behave similarly,

$$\left\|\Sigma^{1/2} R_{S,T}(\Sigma)\right\|^2 \approx \frac{1}{\gamma T}, \qquad \left\|\Sigma^{1/2} R_{0,T}(\Sigma)\right\|^2 \approx \frac{1}{\gamma T},$$

leading to comparable rates. The advantage of tail averaging is again apparent if we consider Assumption (6). In this case for all $r > 0$, if we take $S \propto T$

$$\left\|\Sigma^{1/2} R_{S,T}(\Sigma)\Sigma^r\right\|^2 \approx \left(\frac{1}{\gamma T}\right)^{2r+1}, \tag{10}$$

whereas with uniform averaging one can only prove

$$\left\|\Sigma^{1/2} R_{0,T}(\Sigma)\Sigma^r\right\|^2 \approx \left(\frac{1}{\gamma T}\right)^{2\min(r,1/2)+1}. \tag{11}$$

One immediate observation following from the above discussion is that uniform averaging induces a so-called *saturation effect* [12], meaning that the rates do not improve after $r$ reaches a critical point. As shown above, this effect vanishes considering tail-averaging and the convergence rate of GD is recovered. These results are critically important for our analysis and constitute the main conceptual contribution of our paper. They are proved in Appendix B, while Section A.1 highlights their critical role for SGD. To the best of our knowledge, we are the first to highlight this acceleration property of tail averaging beyond the finite-dimensional setting.

# 4 Main Results and Discussion

In this section we present and discuss our main results. We start by presenting a general bound and then use it to derive the optimal parameter settings and corresponding performance guarantees. A key quantity in our results will be the *effective dimension*

$$\mathcal{N}(1/\gamma L) = \text{Tr}\left[(\Sigma + \frac{1}{\gamma L})^{-1}\Sigma\right] ,$$

introduced in [38] to generalize results from parametric estimation problems to non-parametric kernel methods. Similarly this will be one of the main quantities in our learning bounds.

Further, in all our results we will require that the stepsize is bounded as $\gamma\kappa^2 < 1/4$, and that the tail length $L = T - S$ is scaled appropriately with the total number of iterations $T$. More precisely, our analysis considers two different scenarios where $S = 0$ (plain averaging) is explicitly allowed and where $S > 0$, i.e., where we investigate the merits of tail-averaging. To do so, we will assume $0 \leq S \leq \frac{K-1}{K+1} T$ for some $1 \leq K$, and also $T \leq (K+1)S$ for the latter case.

The following theorem presents a simplified version of our main technical result that we present in its general form in the Appendix. Here, we omit constants and lower order terms for clarity and give the first insights into the interplay between the tuning parameters, namely the step-size $\gamma$, tail-length $L$, and mini-batch size $b$, and the number of points $n$. Note that in a nonparametric setting these are the quantities controlling learning rates. The following result provides a bound for any choice of the tuning parameters, and will allow to derive optimal choices balancing the various error contributions.

**Theorem 1.** *Let $\alpha \in (0, 1]$, $1 \leq L \leq T$ and let Assumption 1 hold. Assume $\gamma\kappa^2 < 1/4$ as well as $n \gtrsim \gamma L \, \mathcal{N}(1/\gamma L)$. Then, the excess risk of the tail-averaged SGD iterates satisfies*

$$\mathbb{E}\left[\left\|\Sigma^{\frac{1}{2}}(\bar{w}_L - w_*)\right\|^2\right] \lesssim \left\|\Sigma^{1/2}R_L(\Sigma)w_*\right\|^2 + \frac{\mathcal{N}(1/\gamma L)}{n} + \frac{\gamma\,\text{Tr}[\Sigma^\alpha]}{b(\gamma L)^{1-\alpha}} .$$

The proof of the result is given in Appendix E. We make a few comments. The first term in the bound is the approximation error, already discussed in Section 3. It is controlled by the bound in (10) and which is decreasing in $\gamma L$. The second term corresponds to a variance error due to sampling and noise in the data. It depends on the effective dimension which is increasing in $\gamma L$. The third term is a computational error due to the randomization in SGD. Note how it depends on both $\gamma L$ and the minibatch size $b$. The larger $b$ is, the smaller this error becomes. The dependence of all three terms on $\gamma L$ suggest already at this stage that $(\gamma L)^{-1}$ plays the role of a regularization parameter. We derive our final bound by balancing all terms, i.e. choosing them to be of the same order. To do so we make additional assumptions. The first one is Eq. (6), enforcing the optimal solution $w_*$ to belong to a subspace of the range of $\Sigma$.

**Assumption 2.** *For some $r \geq 0$ we assume $w_* = \Sigma^r v_*$, for some $v_* \in \mathcal{H}$ satisfying $\|v_*\| \leq R$.*

The larger is $r$ the more stringent is the assumption, or, equivalently, the easier is the problem, see Section 3. A second further assumption is related to the effective dimension.

**Assumption 3.** *For some $\nu \in (0, 1]$ and $C_\nu < \infty$ we assume $\mathcal{N}(1/\gamma L) \leq C_\nu(\gamma L)^\nu$.*

This assumption is common in the nonparametric regression setting, see e.g [6]. Roughly speaking, it quantifies how far $\Sigma$ is from being finite rank. Indeed, it is satisfied if the eigenvalues $(\sigma_i)_i$ of $\Sigma$ have a polynomial decay $\sigma_i \sim i^{-\frac{1}{\nu}}$. Since $\Sigma$ is trace class, the assumption is always satisfied for $\nu = 1$ with $C_\nu = \kappa^2$. Smaller values of $\nu$ lead to faster convergence rates.

The following corollary of Theorem 1, together with Assumptions 2 and 3, derives optimal parameter settings and corresponding learning rates.

**Corollary 1.** *Let all assumptions of Theorem 1 be satisfied, and suppose that Assumptions 2, 3 also hold. Further, assume either*

1. *$0 \leq r \leq 1/2$, $1 \leq L \leq T$ (here $S = 0$, i.e., full averaging is allowed) or*

2. *$1/2 < r$, $1 \leq L < T$ with the additional constraint that for some $K \geq 2$*

$$\frac{K+1}{K-1}S \leq T \leq (K+1)S ,$$

*(only tail-averaging is considered).*

*Then, for any $n$ sufficiently large, the excess risk of the (tail)-averaged SGD iterate satisfies*

$$\mathbb{E}\left[\left\|\Sigma^{\frac{1}{2}}\left(\bar{w}_{L_n} - w_*\right)\right\|^2\right] \lesssim n^{-\frac{2r+1}{2r+1+\nu}}$$

*for each of the following choices:*

(a) $b_n \simeq 1$, $L_n \simeq n$, $\gamma_n \simeq n^{-\frac{2r+\nu}{2r+1+\nu}}$  *(one pass over data)*

(b) $b_n \simeq n^{\frac{2r+\nu}{2r+1+\nu}}$, $L_n \simeq n^{\frac{1}{2r+1+\nu}}$, $\gamma_n \simeq 1$  *(one pass over data)*

(c) $b_n \simeq n$, $L_n \simeq n^{\frac{1}{2r+1+\nu}}$, $\gamma_n \simeq 1$  *($n^{\frac{1}{2r+1+\nu}}$ passes over data)* .

The proof of Corollary 1 is given in Appendix E. It gives optimal rates [6, 5] under different assumptions and choices for the stepsize $\gamma$, the minibatch size $b$ and the tail length $L$, considered as functions of $n$ and the parameters $r$ and $\nu$ from Assumptions 2, 3. We now discuss our findings in more detail and compare them to previous related work.

**Optimality of the bound:**  The above results show that different parameter choices allow to achieve the same error bound. The latter is known to be optimal in minmax sense, see e.g. [6]. As noted before, here we provide simplified statements highlighting the dependence of the bound on the number of points $n$ and the parameters $r$ and $\nu$ that control the regularity of the problem. These are quantities controlling the learning rates and for which lower bounds are available. Note however, that all the constants in the Theorem are worked out and reported in detail in the Appendices.

**Regularization properties of tail-length:**  We recall that for GD it is well known that $(\gamma T)^{-1}$ serves as a regularization parameter, having a quantitatively similar effect to Tikhonov regularization with parameter $\lambda > 0$, see e.g. [12]. More generally, our result shows that in the case of tail averaging the quantity $(\gamma L)^{-1}$ becomes the regularizing parameter for *both GD and SGD*.

**The benefit of tail-averaging:**  For SGD with $b = 1$ and full averaging it has been shown by [10] that a single pass over data (i.e., $T_n = n$) gives optimal rates of convergence provided that $\gamma_n$ is chosen as in case $(a)$ in the corollary. However the results in [10] held only in the case $r \leq 1/2$. Indeed, beyond this regime, there is a saturation effect which precludes optimality for higher smoothness, see the discussion in Section 3, eq. (11). Our analysis for case $(a)$ shows that optimal rates for $r \geq 0$ can still be achieved with the same number of passes and step-size *by using non-trivial tail averaging*. Additionally, we compare our results with those from [26]. In that paper it is shown that multi-passes are beneficial for obtaining improved rates for averaged SGD in a regime where the optimal solution $w^*$ does not belong to $\mathcal{H}$ (Assumption 2 does not hold in that case). In that regime, tail-averaging does not improve convergence. Our analysis focuses on the "opposite" regime where $w^* \in \mathcal{H}$ and saturation slows down the convergence of uniformly-averaged SGD, preventing optimal rates. Here, tail-averaging is indeed beneficial and leads to improved rates.

**The benefit of multi-passes and mini-batching:**  We compare our results with those in [21] where no averaging but mini-batching is considered. In particular, there it is shown that a relatively large stepsize of order $\log(n)^{-1}$ can be chosen provided the minibatch size is set to $n^{\frac{2r+1}{2r+1+\nu}}$ and a number of $n^{\frac{1}{2r+1+\nu}}$ passes is considered. Comparing to these results we can see the benefits of combining minibatching with tail averaging. Indeed from $(c)$ we see that with a comparable number of passes, we can use a larger, constant step-size already with a much smaller minibatch size. Further, comparing $(b)$ and $(c)$ we see that the setting of $\gamma$ and $L$ is the same and there is a full range of possible values for $b_n$ between $[n^{\frac{2r+\nu}{2r+1+\nu}}, n]$ where a constant stepsize is allowed, still ensuring optimality. As noted in [21], increasing the minibatch size beyond a critical value does not yield any benefit. Compared to [21], we show that that tail-averaging can lead to a much smaller critical minibatch size, and hence more efficient computations.

**Comparison to finite-dimensional setting:**  The relationship between the step-size and batch size in finite dimensions dim $\mathcal{H} = d < \infty$ is derived in [19] where also tail-averaging but only one pass over the data is considered. One of the main contributions of this work is characterizing the largest stepsize that allows achieving statistically optimal rates, showing that the largest permissible

stepsize grows linearly in $b$ before hitting a certain quantity $b_{\text{thresh}}$. Setting $b > b_{\text{thresh}}$ results in loss of computational and statistical efficiency: in this regime, each step of minibatch SGD is exactly as effective in decreasing the bias as a step of batch gradient descent. The critical value $b_{\text{thresh}}$ and the corresponding largest admissible stepsize is problem dependent and does not depend on the sample size $n$. Notably, the statistically optimal rate of order $\sigma^2 d/n$ is achieved for all *constant* minibatch sizes, and the particular choice of $b$ only impacts the constants in the decay rate of the bias (which is of the lower order $1/n^2$ anyway). That is, choosing the right minibatch size does not involve a tradeoff between statistical and optimization error. In contrast, our work shows that setting a large batch size $b_n \simeq n^\alpha$, $\alpha \in [0, 1]$ yields optimality guarantees in the infinite dimensional setting. This is due to the fact that choosing the optimal values for parameters like $\gamma$ and $b$ involve a tradeoff between the bias and the variance in this setting. [19] also show that tail-averaging improves the rate at which the initial bias decays if the smallest eigenvalue of the covariance matrix $\sigma_{\min}(\Sigma)$ is lower-bounded by a constant. Their analysis of this algorithmic component is based on observations similar to the ones we made in Section 3. Our analysis significantly extends these arguments by showing the usefulness of tail-averaging in cases when $\sigma_{\min}$ is not necessarily lower-bounded.

## 5 Numerical Illustration

This section provides an empirical illustration to the effects characterized in the previous sections. We focus on two aspects of our results: the benefits of tail-averaging over uniform averaging as a function of the smoothness parameter $r$, and the impact of tail-averaging on the best choice of minibatch sizes. All experiments are conducted on synthetic data with $d = 1,000$ dimensions, generated as follows. We set $\Sigma$ as a diagonal matrix with entries $\Sigma_{ii} = i^{-1/\nu}$ and choose $w^* = \Sigma^r e$, where $e$ is a vector of all 1's. The covariates $X_t$ are generated from a Gaussian distribution with covariance $\Sigma$, and labels are generated as $Y_t = \langle w^*, X_t \rangle + \varepsilon_t$, where $\varepsilon_t$ is standard Gaussian noise. For all experiments, we choose $\nu = 1/2$ and $n = 10,000$. With this choice of parameters, we have seen that increasing $d$ beyond 100 does not yield any noticeable change in the results, indicating that setting $d = 1,000$ is an appropriate approximation to the infinite-dimensional setting.

Our first experiment illustrates the saturation effect described in Section 3 (cf. Eqs. 10,11) by plotting the respective excess risks of uniformly-averaged and tail-averaged SGD as a function of $r$ (Figure 1(a)). We fix $b = 1$ and set $\gamma = n^{-\frac{2r+\nu}{2r+1+\nu}}$ as recommended in Corollary 1. As predicted by our theoretical results, the two algorithms behave similarly for smaller values of $r$, but uniformly-averaged SGD noticeably starts to lag behind its tail-averaged counterpart for larger values of $r$ exceeding $1/2$, eventually flattening out and showing no improvement as $r$ increases. On the other hand, the performance of the tail-averaged version continues to improve for large values of $r$, confirming that this algorithm can indeed massively benefit from favorable structural properties of the data.

In our second experiment, we study the performance of both tail- and uniformly-averaged SGD as a function of the stepsize $\gamma$ and the minibatch-size $b$ (Figure 1(b), (c)). We fix $r = 1/2$ and set $T = n/b$ for all tested values of $b$, amounting to a single pass over the data. Again, as theory predicts, performance remains largely constant as $\gamma \cdot b$ remains constant for both algorithms, until a critical threshold stepsize is reached. However, it is readily apparent from the figures that tail-averaging permits the use of larger minibatch sizes, therefore allowing for more efficient parallelization.

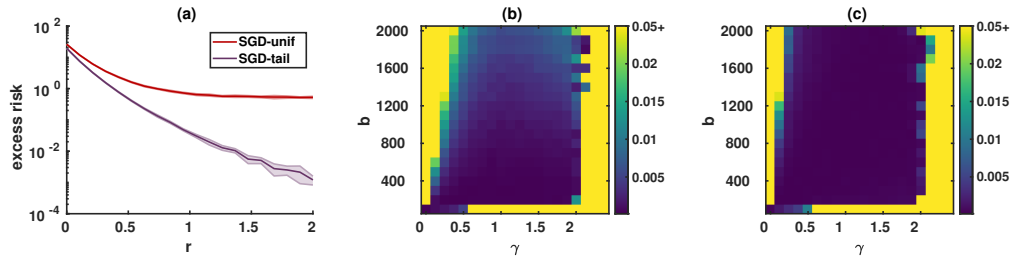

Figure 1: Illustration of the effects of tail-averaging and minibatching. (a) Excess risk as a function of $r$ with uniform and tail averaging. (b) Excess risk as a function of stepsize $\gamma$ and minibatch-size $b$ for SGD with uniform averaging. (c) Excess risk as a function of stepsize $\gamma$ and minibatch-size $b$ for SGD with tail-averaging.

**Acknowledgments**

Nicole Mücke is supported by the German Research Foundation under DFG Grant STE 1074/4-1.
Gergely Neu was supported by *La Caixa* Banking Foundation through the Junior Leader Postdoctoral
Fellowship Program and a Google Faculty Research Award.
Lorenzo Rosasco acknowledges the financial support of the AFOSR projects FA9550-17-1-0390 and
BAA-AFRL-AFOSR-2016-0007 (European Office of Aerospace Research and Development), and
the EU H2020-MSCA-RISE project NoMADS - DLV-777826.

## Footnotes

[1]It is a standard fact that the operator $\Sigma$ is symmetric, positive definite and trace class (hence compact), since $X$ is bounded. Then fractional powers of $\Sigma$ are naturally defined using spectral calculus.

[2]Setting $\frac{d}{ds} s(1 - \gamma s)^T = 0$ gives $1 - \gamma s - s\gamma T = 0 \quad \Rightarrow \quad s = \frac{1}{\gamma(T+1)}$ and $\frac{1}{\gamma(T+1)} \left( 1 - \gamma \frac{1}{\gamma(T+1)} \right)^t \le \frac{1}{\gamma t}$.

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
