[Supplementary Material · cam-ready-6831-supp.pdf]

# Appendix: Beating SGD Saturation with Tail-Averaging and Minibatching

## A   Analysis

This section presents the key components of the proofs of our main results. Recall that the goal of the analysis is to understand the rate at which the tail-averaged SGD iterates $\bar{w}_{S,T}$ approach the risk minimizer $w_*$. The main error decomposition underlying our proofs is borrowed from [21], and based on introducing two intermediate objects that will be shown to converge towards $w_*$, yet stay close to the SGD iterates $w_t$. In Section 3 we have already introduced one of these components: population GD. We will further need the *empirical* (batch) GD iteration, defined as

$$v_{t+1} = v_t - \gamma \, \frac{1}{n} \sum_{j=1}^{n} \left( \langle v_t, x_j \rangle_{\mathcal{H}} - y_j \right) x_j = \left( I - \gamma \widehat{\Sigma} \right) v_t - \gamma \hat{h}, \tag{12}$$

where we also introduced the important notations

$$\widehat{\Sigma} = \frac{1}{n} \sum_{j=1}^{n} x_j \otimes x_j \qquad\qquad \hat{h} = \frac{1}{n} \sum_{j=1}^{n} x_j y_j.$$

Analogously to the tail-averaged SGD/GD we define the tail-averaged batch GD iterates

$$\bar{v}_{S,T} = \frac{1}{T-S} \sum_{t=S+1}^{T} v_t \, ,$$

which will act as our proxy to $\bar{w}_{S,T}$. With these definitions in place, we can upper bound the excess risk of $\bar{w}_{S,T}$ as

$$\left\| \Sigma^{1/2}(w_* - \bar{w}_{S,T}) \right\|^2 \leq 2 \left\| \Sigma^{1/2}(w_* - \bar{v}_{S,T}) \right\|^2 + 2 \left\| \Sigma^{1/2}(\bar{v}_{S,T} - \bar{w}_{S,T}) \right\|^2. \tag{13}$$

The purpose of this decomposition is to help us separate the inherent statistical errors due to using an i.i.d. sample of fixed size $n$ (first term) and the errors introduced by the randomized algorithm (second term). Accordingly, we will refer to this latter term as the *computational variance*. In the sections below, we give bounds on both terms separately.

### A.1   Learning properties of GD with tail averaging

In this section, we discuss how to bound the first term in the decomposition of Equation (13). In analogy to the discussion in Section 3, we rewrite the empirical GD using spectral filtering functions,

$$v_{t+1} = g_{t+1}\big(\widehat{\Sigma}\big)\hat{h}, \qquad\qquad g_{t+1}\big(\widehat{\Sigma}\big) = \gamma \sum_{j=0}^{t} \big(I - \gamma\widehat{\Sigma}\big)^{j} \tag{14}$$

With this notation, the tail-averaged GD iterates can be written as

$$\bar{v}_{T,S} = G_{S,T}(\widehat{\Sigma})\hat{h} \, , \qquad\qquad G_{S,T}(\sigma) = \frac{1}{T-S} \sum_{t=S+1}^{T} g_t(\sigma). \tag{15}$$

Most of the analysis in this section will rely on the regularization properties of the spectral filter $G_{S,T}(\widehat{\Sigma})$, the corresponding residual operators

$$R_{S,T}(\widehat{\Sigma}) = 1 - \widehat{\Sigma}G_{S,T}(\widehat{\Sigma}), \tag{16}$$

and the analogous population quantities introduced in Section 3. Denoting the tail-length by $L = T - S$, we will occasionally use the notations $G_L = G_{S,T}$ and $R_L = R_{S,T}$.

Our error bounds are derived by means of a classical error decomposition in bias and variance (see, e.g., 6, 3, 5 and 22). Recalling the definition of the averaged population GD in Equations (7) and (8),

we consider the decomposition

$$
\begin{aligned}
\bar{v}_L - w_* &= (\bar{v}_L - \bar{u}_L) + (\bar{u}_L - w_*) = (\bar{v}_L - \bar{u}_L) + R_L(\Sigma)w_* \\
&= \left(G_L(\widehat{\Sigma})\hat{h} - G_L(\widehat{\Sigma})\widehat{\Sigma}\bar{u}_L\right) + \left(G_L(\widehat{\Sigma})\widehat{\Sigma}\bar{u}_L - \bar{u}_L\right) + R_L(\Sigma)w_* \\
&= G_L(\widehat{\Sigma})\left(\hat{h} - \widehat{\Sigma}\bar{u}_L\right) + R_L(\widehat{\Sigma})\bar{u}_L + R_L(\Sigma)w_* \; .
\end{aligned}
\tag{17}
$$

We refer to

$$
\mathcal{A}(L) = \left\|\Sigma^{1/2}R_L(\Sigma)w_*\right\|^2
\tag{18}
$$

as the *deterministic approximation error*, to

$$
\hat{\mathcal{A}}(L) = \left\|\Sigma^{1/2}R_L(\hat{\Sigma})\bar{u}_L\right\|^2
\tag{19}
$$

as the *stochastic approximation error* and to

$$
\widehat{\mathcal{V}}(L) = \left\|\Sigma^{1/2}G_L(\hat{\Sigma})(\hat{h} - \widehat{\Sigma}\bar{u}_L)\right\|^2
\tag{20}
$$

as the *sample variance*. Our analysis will crucially rely on the properties of the residual operator $R_{S,T}$ already discussed in Section 3. Here we show that these arguments made about population GD also impacts the learning error for empirical GD in the same qualitative way. More precisely, Propositions 1 and 2 in Appendices C.1 and C.2 show that, under appropriate conditions, the (expected) approximation errors can be bounded as

$$
\mathcal{A}(L) \lesssim \begin{cases} R^2(\gamma L)^{-2(r+1/2)} & \text{if } r \leq \frac{1}{2}, \\ K^2 R^2(\gamma L)^{-2(r+1/2)} & \text{else, and} \end{cases} \qquad \mathbb{E}\left[\hat{\mathcal{A}}(L)\right] \lesssim \begin{cases} R^2(\gamma L)^{-2(r+1/2)} & \text{if } r \leq \frac{1}{2}, \\ K^{4(r+1)} R^2(\gamma L)^{-2(r+1/2)} & \text{else.} \end{cases}
$$

Notably, proving this result for $r > 1/2$ critically relies on setting $S$ as a *constant* fraction of $T$ that enables the rapid decay of $R_{S,T}$ in $S$, highlighting the important role of tail averaging to obtain these results. The precise condition we require is $S \leq \frac{K-1}{K+1}T$ and $T \leq (K+1)T$ to hold for some constant $K > 1$ see Corollary 1. Regarding the sample variance, Proposition 4 in Appendix C.3 shows the bound

$$
\mathbb{E}\left[\hat{\mathcal{V}}(L)\right] \lesssim \mathcal{A}(L) + \frac{\gamma L(1 + \|w_*\|^2)}{n^2} + \frac{\mathcal{N}(1/\gamma L)}{n}.
$$

Putting these results together, we can conclude that the excess risk of tail-averaged GD satisfies the bound

$$
\mathbb{E}\left[\left\|\Sigma^{1/2}(w_* - \bar{v}_{S,T})\right\|^2\right] \lesssim R^2(\gamma L)^{-2(r+1/2)} + \frac{\gamma L(1 + \|w_*\|^2)}{n^2} + \frac{\mathcal{N}(1/\gamma L)}{n}
$$

whenever $K$ is set as $O(1)$. The precise bound is stated in Appendix C.4 as Theorem 2. A particularly important consequence of this result is that, under the additional Assumption 3, the excess-risk bound can be further rewritten as

$$
\mathbb{E}\left[\left\|\Sigma^{1/2}(\bar{v}_L - w_*)\right\|^2\right] \leq R^2\, C_K\, n^{-\frac{2r}{2r+1+\nu}} \; ,
$$

when choosing $\gamma_n \simeq n^{-a}$ and $T \simeq n^{\tilde{a}}$ for some $a, \tilde{a} > 0$ satisfying $a - \tilde{a} = \frac{1}{2r+1+\nu}$. Once again, these results rely on choosing $T \simeq S$ in the case $r > 1/2$, whereas choosing $S = 0$ is sufficient for the case $r \leq 1/2$. This result is formally stated as Corollary 2 in Appendix C.4.

In the low smoothness regime, i.e. $0 \leq r \leq 1/2$, the choice $0 < S, S_n \asymp T_n$ is also possible but does not affect the rate of convergence, whereas in the high smoothness regime, i.e. $1/2 < r$, tail averaging is necessary to avoid saturation.

## A.2 SGD vs. GD

We now move on to analyze the difference between the tail-averaged SGD and GD iterates and provide a bound on the second term in the decomposition (13). To relate the two iterations, we introduce the notation

$$
\widehat{\Sigma}_t = \frac{1}{b}\sum_{i=b(t-1)+1}^{bt} x_{j_i} \otimes x_{j_i} \quad \text{and} \quad \hat{h}_t = \frac{1}{b}\sum_{i=b(t-1)+1}^{bt} y_{j_i} x_{j_i},
$$

so that the minibatch SGD iteration can be written as $w_{t+1} = (I - \gamma\widehat{\Sigma}_t)w_t - \gamma\hat{h}_t$. Thus, the difference between the two iterate sequences can be written in the recursive form

$$w_{t+1} - v_{t+1} = (I - \gamma\widehat{\Sigma}_{t+1})(w_t - v_t) + \gamma\xi_{t+1}, \tag{21}$$

where $\xi_{t+1} = \xi_{t+1}^{(1)} + \xi_{t+1}^{(2)}$ and

$$\xi_{t+1}^{(1)} = (\hat{\Sigma} - \hat{\Sigma}_{t+1})v_t, \quad \xi_{t+1}^{(2)} = \hat{h}_{t+1} - \hat{h}. \tag{22}$$

It is easy to see that $\xi_{t+1}$ has zero mean when conditioned on the history $\mathcal{F}_t$ and the dataset. Notice that the recursion above is of the form

$$\mu_{t+1} = (I - \gamma\widehat{H}_{t+1})\mu_t + \gamma\zeta_{t+1}$$

for i.i.d. self-adjoint positive operators operators $\widehat{H}_t$ satisfying $\mathbb{E}[\widehat{H}_t|\mathcal{F}_t] = H$ and $\mathbb{E}[\zeta_t|\mathcal{F}_t] = 0$. Such recursions have been well studied in the stochastic approximation literature, and can be analyzed by techniques proposed by [1] (and later used by 2, 10, 26, among many others). Our analysis builds on a recent result by [26] that we generalize to account for minibatching and tail-averaging. On a high level, this result states that if $\widehat{H}_t$ and $\zeta_t$ respectively satisfy $\mathbb{E}[\widehat{H}_t^2|\mathcal{F}_t] \preccurlyeq \kappa^2 H$ and $\mathbb{E}[\zeta_t \otimes \zeta_t|\mathcal{F}_t] \preccurlyeq \sigma^2\widehat{\Sigma}$ for some $\kappa,\sigma$, then the tail-averaged iterate $\bar{\mu}_{S,T} = \frac{1}{T-S}\sum_{t=S+1}^{T}\mu_t$ satisfies

$$\mathbb{E}\left[\left\|H^{\frac{u}{2}}\bar{\mu}_{S,T}\right\|^2\right] \lesssim \sigma^2\operatorname{Tr}[H^\alpha]\gamma^{1-u+\alpha}(T-S)^{\alpha-u}\frac{S+1}{T-S}$$

for arbitrary $\alpha \in (0,1]$ and $u \in [0, 1+\alpha]$. Appendix D is dedicated to formally proving this result, presented precisely as Proposition 5.

Our analysis crucially relies on applying the above lemma for $H = \widehat{\Sigma}$ under an appropriately defined condition $\mathcal{E}_1$ on the data, see (68), guaranteeing that $\widehat{\Sigma}$ is "close enough" to its population counterpart $\Sigma$. A second condition $\mathcal{E}_2$ in (69) ensures the boundedness of $\mathbb{E}[\xi_t \otimes \xi_t|\mathcal{F}_t, \mathcal{G}_n]$, conditioned on the data $\mathcal{G}_n$. We note that, ensuring the condition about $\xi_t$ is rather challenging due to the fact that the size of $\xi_t^{(2)}$ depends on the norm of the GD iterate $\|v_t\|$, which can be, in principle, unbounded. Consequently, the resulting error terms can only be controlled in a probabilistic sense. Our analysis relies on showing that there indeed exists a condition $\mathcal{E}_1 \cap \mathcal{E}_2$ that holds with high probability and ensures the desired properties. A formal treatment of these matters is presented in Appendix E. The final result of these derivations is Proposition 6 that, under appropriate conditions on the algorithm's parameters, bounds the deviations between the averaged GD and SGD iterates as

$$\mathbb{E}\left[\left\|\Sigma^{\frac{1}{2}}(\bar{w}_{S,T} - \bar{v}_{S,T})\right\|^2\right] \lesssim \frac{\gamma^\alpha\operatorname{Tr}[\Sigma^\alpha]}{bL^{1-\alpha}}.$$

## B  Spectral Filtering properties of averaged GD

Consider the function

$$g_t(\sigma) = \gamma\sum_{k=0}^{t-1}(1-\gamma\sigma)^k = \sigma^{-1}(1 - (1-\gamma\sigma)^t). \tag{23}$$

defined on the spectrum $\sigma(\Sigma) \subseteq [0, \kappa^2]$ of $\Sigma$ and let

$$r_t(\sigma) = 1 - \sigma g_t(\sigma).$$

Then, for any $\alpha \in [0,1]$ [12]

$$\sup_{0<\sigma\leq\kappa^2}|\sigma^\alpha g_t(\sigma)| \leq (\gamma t)^{1-\alpha}. \tag{24}$$

Moreover, for any $0 \leq u$

$$\sup_{0<\sigma\leq\kappa^2}|r_t(\sigma)|\sigma^u \leq C_u(\gamma t)^{-u}, \tag{25}$$

for some $C_u > 0$. In particular, $C_0 = 1$.

For $0 \leq S \leq T-1$ consider

$$G_{S,T}(\sigma) = \frac{1}{T-S}\sum_{t=S+1}^{T}g_t(\sigma)$$

and let
$$R_{T,S}(\sigma) = 1 - \sigma G_{S,T}(\sigma).$$

**Lemma 1** (Filter).

$$\sigma G_{S,T}(\sigma) = 1 - \frac{1}{(T-S)\gamma\sigma}(1-\gamma\sigma)^{S+1}\big(1 - (1-\gamma\sigma)^{T-S}\big) .$$

*Proof of Lemma 1.* By (23), we have

$$\sigma G_{S,T}(\sigma) = \frac{1}{T-S}\sum_{t=S+1}^{T} 1 - (1-\gamma\sigma)^t$$

$$= 1 - \frac{1}{T-S}\sum_{t=S+1}^{T}(1-\gamma\sigma)^t$$

$$= 1 - \frac{1}{T-S}\sum_{t=0}^{T}(1-\gamma\sigma)^t + \frac{1}{T-S}\sum_{t=0}^{S}(1-\gamma\sigma)^t$$

$$= 1 - \frac{\big((1-\gamma\sigma)^{S+1} - (1-\gamma\sigma)^{T+1}\big)}{(T-S)\gamma\sigma}$$

$$= 1 - \frac{(1-\gamma\sigma)^{S+1}(1 - (1-\gamma\sigma)^{T-S})}{(T-S)\gamma\sigma} .$$

$\square$

**Lemma 2** (Properties I). *Let $u \in [0,1]$. For any $1 \leq T, 0 \leq S \leq T-1$ we have*

$$\sup_{0<\sigma\leq\kappa^2} |\sigma^u G_{S,T}(\sigma)| \leq C_u\, \gamma^{1-u}\, (T+S)(T-S)^{-u} .$$

*In particular, for $1 \leq K$, choosing $S \leq \frac{K-1}{K+1}\,T$ gives*

$$\sup_{0<\sigma\leq\kappa^2} |\sigma^u G_{S,T}(\sigma)| \leq K\gamma^{1-u}(T-S)^{1-u} .$$

*Proof.* By (16) we have

$$\sup_{0<\sigma\leq\kappa^2} |\sigma^u G_{S,T}(\sigma)| \leq \sup_{0<\sigma\leq\kappa^2} \frac{1}{T-S}\sum_{t=S+1}^{T} |\sigma^u g_t(\sigma)|$$

$$\leq \frac{\gamma^{1-u}}{T-S}\sum_{t=S+1}^{T} t^{1-u} .$$

From Lemma 14 and Lemma 12, we find

$$\sum_{t=S+1}^{T} t^{1-u} \leq \int_{S+1}^{T} t^{1-u}\, dt$$

$$= \frac{1}{2-u}\big(T^{2-u} - (S+1)^{2-u}\big)$$

$$\leq \frac{1}{2-u}\big(T^{2-u} - S^{2-u}\big)$$

$$\leq \frac{1}{2-u}(T+S)(T-S)^{1-u} .$$

This proves the first statement. The second statement follows by observing that $T+S \leq K(T-S)$ if $S \leq \frac{K-1}{K+1}\,T$.

$\square$

**Remark 1.** *A more refined bound for the case $u = 1$ can be obtained by considering (16), which directly leads to*

$$\sup_{0 < \sigma \le \kappa^2} |\sigma G_{S,T}(\sigma)| \le \sup_{0 < \sigma \le \kappa^2} \frac{1}{T - S} \sum_{t=S+1}^{T} |\sigma g_t(\sigma)| \le 1 .$$

**Lemma 3** (Properties II)**.** *Let $1 \le T$ and $0 \le S \le T - 1$.*

1. *For any $u \in [0, 1]$, we have*

$$\sup_{0 < \sigma \le \kappa^2} |\sigma^u R_{S,T}(\sigma)| \le \gamma^{-u} (T - S)^{-u} .$$

2. *For any $u > 1$ we have*

$$\sup_{0 < \sigma \le \kappa^2} |\sigma^u R_{S,T}(\sigma)| \le \tilde{C}_u \, \gamma^{-u} \, \frac{S+1}{T-S} \left( \frac{1}{S+1} \right)^u ,$$

*for some $\tilde{C}_u < \infty$. In particular, if $1 \le K$ and $S \le \frac{K-1}{K+1} T$ one has*

$$\sup_{0 < \sigma \le \kappa^2} |\sigma^u R_{S,T}(\sigma)| \le \tilde{C}_u \, \gamma^{-u} \, K \left( \frac{1}{S+1} \right)^u .$$

*If additionally $T \le (K + 1)S$, one has*

$$\sup_{0 < \sigma \le \kappa^2} |\sigma^u R_{S,T}(\sigma)| \le 2\tilde{C}_u \, \gamma^{-u} \, K^2 \left( \frac{1}{T-S} \right)^u .$$

*Proof of Lemma 3.* By Lemma 1 we have for any $u \in [0, 1]$

$$
\begin{aligned}
|\sigma^u R_{S,T}(\sigma)| &= \frac{\sigma^u}{(T-S)\gamma\sigma}(1 - \gamma\sigma)^{S+1}(1 - (1 - \gamma\sigma)^{T-S}) \\
&\le \frac{\sigma^u}{(T-S)\gamma\sigma}(1 - \gamma\sigma)^{S+1}((T-S)\gamma\sigma)^{1-u} \\
&= (\gamma(T-S))^{-u}(1 - \gamma\sigma)^{S+1} \\
&\le \gamma^{-u}(T - S)^{-u} ,
\end{aligned}
$$

where we use that

$$|1 - (1 - x)^t| \le (tx)^{1-u}$$

for any $x \in [0, 1]$ and for any $u \in [0, 1]$.

For $u > 1$ we apply (25) and Lemma 14 and obtain

$$
\begin{aligned}
\sup_{0 < \sigma \le \kappa^2} |\sigma^u R_{S,T}(\sigma)| &\le C_u \frac{\gamma^{-u}}{T-S} \sum_{t=S+1}^{T} t^{-u} \\
&\le C_u \frac{\gamma^{-u}}{T-S} \left( (S+1)^{-u} + \int_{S+1}^{T} t^{-u} dt \right) \\
&\le C_u \frac{\gamma^{-u}}{T-S} \left( (S+1)^{-u} + \frac{1}{u-1} \left( \left( \frac{1}{S+1} \right)^{u-1} - \left( \frac{1}{T+1} \right)^{u-1} \right) \right) \\
&\le \tilde{C}_u \, \gamma^{-u} \, \frac{S+1}{T-S} \left( \frac{1}{S+1} \right)^u .
\end{aligned}
\tag{26}
$$

Note that $S \le \frac{K-1}{K+1} T$ implies

$$\frac{S+1}{T-S} \le K .$$

Finally, $T \le (K + 1)S$ gives

$$\frac{1}{S+1} \le \frac{K+1}{T-S} \le \frac{2K}{T-S} .$$

$\square$

# C   Bounds Tail-Averaged Gradient Descent

Our error bounds are derived by means of a classical error decomposition in bias and variance, see e.g. [6], [3], [5] and [22]. More precisely, recalling the filter expression of the population GD,

$$u_L = G_L(\Sigma)\Sigma w_* , \tag{27}$$

we consider

$$
\begin{aligned}
\bar{v}_L - w_* &= (\bar{v}_L - u_L) + (u_L - w_*) \\
&= (\bar{v}_L - u_L) + R_L(\Sigma)w_* \\
&= (G_L(\hat{\Sigma})\hat{h} - G_L(\hat{\Sigma})\hat{\Sigma}u_L) + (G_L(\hat{\Sigma})\hat{\Sigma}u_L - u_L) + R_L(\Sigma)w_* \\
&= G_L(\hat{\Sigma})(\hat{h} - \hat{\Sigma}u_L) + R_L(\hat{\Sigma})u_L + R_L(\Sigma)w_* .
\end{aligned}
\tag{28}
$$

We refer to

$$\mathcal{A}(L) = ||\Sigma^{1/2}R_L(\Sigma)w_*||^2 \tag{29}$$

as the *deterministic Approximation error*, to

$$\hat{\mathcal{A}}(L) = ||\Sigma^{1/2}R_L(\hat{\Sigma})u_L||^2 \tag{30}$$

as the *stochastic Approximation error* and to

$$\widehat{\mathcal{V}}(L) = ||\Sigma^{1/2}G_L(\hat{\Sigma})(\hat{h} - \hat{\Sigma}u_L)||^2 \tag{31}$$

as the *Sample variance*. In what follows we successively bound each error term in Section C.1, Section C.2 and Section C.3. Finally, the total bound is given in Section C.4.
In the following we let

$$\Sigma_L = (\Sigma + \frac{1}{\gamma L}), \qquad \text{and} \qquad \hat{\Sigma}_L = (\hat{\Sigma} + \frac{1}{\gamma L}).$$

## C.1   Bounding the deterministic Approximation Error

**Proposition 1** (Deterministic Approximation Error). *Let $1 \leq T$, $0 \leq S \leq T - 1$, $\gamma\kappa^2 < 1$ and Assumption 2 hold.*

   *1. If $0 \leq r \leq 1/2$, we have*

$$\mathcal{A}(L) \leq R^2 \, (\gamma L)^{-2(r+1/2)} .$$

   *2. If $1/2 < r$ we have*

$$\mathcal{A}(L) \leq C_r \, R^2 \, \gamma^{-2(r+1/2)} \left(\frac{S+1}{L}\right)^2 \left(\frac{1}{S+1}\right)^{2(r+1/2)} .$$

   *for some $C_r < \infty$. In particular, if $1 \leq K$, $S \leq \frac{K-1}{K+1} T$ and $T \leq (K+1)S$, one has*

$$\mathcal{A}(L) \leq C_r \, K^2 \, R^2 \, (\gamma L)^{-2(r+1/2)} .$$

*Proof of Proposition 1.* By Assumption 2 we have

$$\mathcal{A}(L) = ||\Sigma^{1/2}R_L(\Sigma)w_*||^2 \leq R^2 \, ||\Sigma^{r+1/2}R_L(\Sigma)||^2 .$$

Since

$$||\Sigma^{r+1/2}R_L(\Sigma)|| \leq \sup_{0 < \sigma \leq \kappa^2} |\sigma^{r+1/2}R_L(\sigma)| ,$$

the result follows immediately by applying Lemma 3. $\qquad\square$

## C.2  Bounding the stochastic Approximation Error

**Proposition 2** (Stochastic Approximation Error). *Let $1 \leq T$, $0 \leq S \leq T - 1$, $\gamma\kappa^2 < 1$ and Assumption 2 hold. Further assume*

$$n \geq 16\kappa^2 \ \gamma L \ \max\{1, \mathcal{N}(1/(\gamma L))\} \ .$$

*1. If $0 \leq r \leq 1/2$, we have*

$$\mathbb{E}\left[\hat{\mathcal{A}}(L)\right] \leq C_r R^2 \left(\frac{T+S}{L}\right)^2 (\gamma L)^{-2(r+1/2)} \ ,$$

*for some $C_r < \infty$.*

*2. If $1/2 < r$ we have*

$$\mathbb{E}\left[\hat{\mathcal{A}}(L)\right] \leq C_{r,\kappa} R^2 \left(\frac{T+S}{L}\right)^2 \left((\gamma L) \frac{(S+1)^2}{L^2} \ \Psi_r^2(S,T) + \frac{1}{n}\right) \ ,$$

*for some $C_{r,\kappa} < \infty$ and where $\Psi_r$ is defined in (37). In particular, if $1 \leq K$, $S \leq \frac{K-1}{K+1} T$ and $T \leq (K+1)S$, one has*

$$\mathbb{E}\left[\hat{\mathcal{A}}(L)\right] \leq \tilde{C}_{r,\kappa} \ K^{4(r+1)}(\gamma L)^{-2(r+1/2)} \ ,$$

*for some $\tilde{C}_{r,\kappa} < \infty$.*

*Proof of Proposition 2.* We start with deriving bounds holding with high probability, bounds in expectation follow then by integration. From Lemma 10 we derive with probability at least $1 - \delta/2$

$$\hat{\mathcal{A}}(L) = ||\Sigma^{1/2} R_L(\hat{\Sigma}) u_L||^2$$
$$\leq 16 \log^2(4\delta^{-1}) ||\hat{\Sigma}_L^{1/2} R_L(\hat{\Sigma}) u_L||^2 \ . \tag{32}$$

We separate the analysis by considering two cases.

**Case 1** ($0 \leq r \leq 1/2$)**:** Recalling the definition of $u_L$ in (27) gives

$$||\hat{\Sigma}_L^{1/2} R_L(\hat{\Sigma}) u_L|| \leq R \ ||\hat{\Sigma}_L R_L(\hat{\Sigma})|| \cdot ||\hat{\Sigma}_L^{-1/2} \Sigma_L^{1/2}|| \cdot ||\Sigma^{r+1/2} G_L(\Sigma)|| \ .$$

Bounding the first term is done by using Lemma 1 and Lemma 3, leading to

$$||\hat{\Sigma}_L R_L(\hat{\Sigma})|| \leq \sup_{0 < \sigma \leq \kappa^2} |(\sigma + 1/(\gamma L)) R_L(\sigma)|$$
$$\leq \sup_{0 < \sigma \leq \kappa^2} |\sigma R_L(\sigma)| + 1/(\gamma L) \sup_{0 < \sigma \leq \kappa^2} |R_L(\sigma)|$$
$$\leq 2(\gamma L)^{-1} \ .$$

From Lemma 2 we obtain

$$||\Sigma^{r+1/2} G_L(\Sigma)|| \leq C_r \ \gamma^{1/2-r} \ (T+S) L^{-(r+1/2)} \ .$$

Thus, applying Corollary 3, gives with probability at least $1 - \delta/2$

$$\hat{\mathcal{A}}(L) \leq 64 \cdot 16 C_r^2 R^2 \log^4(4\delta^{-1}) \ (\gamma L)^{-2} \ \gamma^{1-2r} \ (T+S)^2 L^{-2(r+1/2)} \ , \tag{33}$$

for some $C_r < \infty$.

**Case 2** ($1/2 < r$)**:** In this case we split (32) differently. Using Assumption 2 and Definition (27), we obtain

$$||\hat{\Sigma}_L^{1/2} R_L(\hat{\Sigma}) u_L|| = ||\hat{\Sigma}_L^{1/2} R_L(\hat{\Sigma}) G_L(\Sigma) \Sigma w_*||$$
$$\leq R|| \underbrace{\hat{\Sigma}_L^{1/2} R_L(\hat{\Sigma}) (\Sigma^{r+1/2} - \hat{\Sigma}^{r+1/2}) G_L(\Sigma) \Sigma^{1/2}}_{A_1} ||$$
$$+ \ R|| \underbrace{\hat{\Sigma}_L^{1/2} R_L(\hat{\Sigma}) \hat{\Sigma}^{r+1/2} G_L(\Sigma) \Sigma^{1/2}}_{A_2} || \ . \tag{34}$$

**Bounding $A_1$:** For bounding $A_1$ we apply [5], Proposition 5.5. and Proposition 5.6., to obtain with probability at least $1 - \delta/2$

$$||\Sigma^{r+1/2} - \hat{\Sigma}^{r+1/2}|| \leq C_r ||\Sigma - \hat{\Sigma}||$$

$$\leq 6C_r \frac{\kappa^2}{\sqrt{n}} \log(4\delta^{-1}) .$$

Furthermore, Lemma 2 gives

$$||G_L(\Sigma)\Sigma^{1/2}|| \leq C \, \gamma^{1/2} \, (T + S)L^{-1/2} , \tag{35}$$

for some numerical constant $C < \infty$. Moreover, using Lemma 3 leads to

$$||\hat{\Sigma}_L^{1/2} R_L(\hat{\Sigma})|| \leq \sup_{0<\sigma\leq\kappa^2} |(\sigma + (1/(\gamma L)))^{1/2} R_L(\sigma)|$$

$$\leq \sup_{0<\sigma\leq\kappa^2} |\sigma^{1/2} R_L(\sigma)| + (1/(\gamma L))^{1/2} \sup_{0<\sigma\leq\kappa^2} |R_L(\sigma)|$$

$$\leq 2(\gamma L)^{-1/2} .$$

Collecting the previous steps we arrive at

$$||A_1|| \leq C_r' \left(\frac{T+S}{L}\right) \frac{\kappa^2}{\sqrt{n}} \log(4\delta^{-1}) , \tag{36}$$

with probability at least $1 - \delta/2$, for some numerical constant $C_r' < \infty$.

**Bounding $A_2$:** For bounding $A_2$ we apply Lemma 3 once more, giving

$$||\hat{\Sigma}_L^{1/2} R_L(\hat{\Sigma})\hat{\Sigma}^{r+1/2}||$$

$$\leq \sup_{0<\sigma\leq\kappa^2} |(\sigma + (1/(\gamma L)))^{1/2} R_L(\sigma)\sigma^{r+1/2}|$$

$$\leq \sup_{0<\sigma\leq\kappa^2} |\sigma^{r+1} R_L(\sigma)| + (\gamma L)^{-1/2} \sup_{0<\sigma\leq\kappa^2} |\sigma^{r+1/2} R_L(\sigma)|$$

$$\leq C_r' \, \gamma^{-(r+1)} \frac{S+1}{L} \left(\frac{1}{S+1}\right)^{r+1} + C_r'' \, \gamma^{-(r+1/2)} \frac{S+1}{L} \left(\frac{1}{S+1}\right)^{r+1/2} (\gamma L)^{-1/2}$$

$$\leq C_r''' \frac{S+1}{L} \, \Psi_r(S,T) .$$

where we set

$$\Psi_r(S,T) := \gamma^{-(r+1)} \left[\left(\frac{1}{S+1}\right)^{r+1} + L^{-1/2}\left(\frac{1}{S+1}\right)^{r+1/2}\right] . \tag{37}$$

Thus, combining with (35), we find

$$||A_2|| \leq \tilde{C}_r \, (\gamma L)^{1/2} \frac{T+S}{L} \frac{S+1}{L} \, \Psi_r(S,T) . \tag{38}$$

Finally, note that $S \leq \frac{K-1}{K+1} T$ implies

$$\frac{T+S}{L} \leq K , \qquad \frac{S+1}{L} \leq K$$

and $T \leq (K+1)S$ gives

$$\frac{1}{S+1} \leq \frac{K+1}{L} \leq \frac{2K}{L} .$$

Hence,

$$\Psi_r(S,T) \leq (4K)^{r+1}(\gamma L)^{-(r+1)} . \tag{39}$$

Thus,

$$||A_2|| \leq \tilde{C}_r \, (4K)^{2(r+1)}(\gamma L)^{-(r+1/2)} . \tag{40}$$

The result in this case then follows by combining (39), (34) with (36), (38) and (40) and by integration, Lemma 11 . $\qquad\square$

## C.3 Bounding the Sample Variance

For proving the bound for the sample variance we need a concentration result which we slightly generalize from [22].

**Proposition 3.** *Let $u_L$ be defined by (27), $\mathcal{A}(L)$ by (18) and $\delta \in (0,1]$. Under Assumption 1, one has with probability at least $1 - \delta$*

$$\left\| (\Sigma + \frac{1}{\gamma L})^{-1/2} \Big( (\hat{\Sigma} u_L - \hat{h}) - (\Sigma u_L - h) \Big) \right\|$$

$$\leq c \log(2\delta^{-1}) \left( \frac{\sqrt{\gamma L}(\kappa M + \kappa^2 \|u_L\|)}{n} + \sqrt{\frac{\kappa^2 \gamma L \mathcal{A}(L) + \mathcal{N}(\frac{1}{\gamma L})}{n}} \right) ,$$

*for some numerical constant $c < \infty$.*

**Proposition 4** (Sample Variance). *Set $L = T - S$ and assume $\gamma \kappa^2 < 1$ as well as*

$$n \geq 16\kappa^2 \, \gamma L \, \max\{1, \mathcal{N}(1/(\gamma L))\} \ .$$

*Under Assumption 1 one has*

$$\mathbb{E}\Big[ \hat{\mathcal{V}}(L) \Big] \leq C_{\tilde{\sigma}, M, \kappa} \left( 1 + \frac{\Delta(S,T)}{L^2} \right)^2$$
$$\left( \mathcal{A}(L) + \frac{\gamma L(1 + \|w_*\|^2)}{n^2} + \frac{\mathcal{N}(1/\gamma L)}{n} \right)$$

*for some $C_{\tilde{\sigma}, M, \kappa} < \infty$ and where*

$$\Delta(S,T) = T(T+1) - S(S+1) \ .$$

*In particular, if $1 \leq K$ and $S \leq \frac{K-1}{K+1} T$ one has*

$$1 + \frac{\Delta(S,T)}{L^2} \leq 1 + 2K \ .$$

*Proof of Proposition 4.* According to Lemma 10, we have with probability at least $1 - \delta/2$

$$\hat{\mathcal{V}}(L) = \|\Sigma^{1/2} G_L(\hat{\Sigma})(\hat{h} - \hat{\Sigma} u_L)\|^2$$
$$\leq 16 \log^2(4\delta^{-1}) \, \|\hat{\Sigma}_L^{1/2} G_L(\hat{\Sigma})(\hat{h} - \hat{\Sigma} u_L)\|^2 \ . \tag{41}$$

We proceed by decomposing as follows:

$$\hat{\Sigma}_L^{\frac{1}{2}} G_L(\hat{\Sigma})(\hat{h} - \hat{\Sigma} u_L) = \hat{\Sigma}_L G_L(\hat{\Sigma}) \cdot \hat{\Sigma}_L^{-1/2} \Sigma_L^{1/2} \cdot \hat{h}_L \ ,$$

with

$$\hat{h}_L = \Sigma_L^{1/2} \Big( \hat{h} - \hat{\Sigma} u_L \Big) \ .$$

Using the filter function properties in Lemma 1 and Lemma 2 gives

$$\|\hat{\Sigma}_L G_L(\hat{\Sigma})\| \leq \sup_{0 < \sigma \leq \kappa^2} |(\sigma + 1/(\gamma L)) G_L(\sigma)|$$
$$\leq 1 + \frac{1}{L^2} (T(T+1) - S(S+1))$$
$$= 1 + \frac{\Delta(S,T)}{L^2} \ , \tag{42}$$

with $\Delta(S,T) = T(T+1) - S(S+1)$. Furthermore, Corollary 3 gives

$$\|\hat{\Sigma}_L^{-1/2} \Sigma_L^{1/2}\| \leq 4 \log(8\delta^{-1}) \tag{43}$$

with probability at least $1 - \delta/4$.

For bounding $\hat{h}_L$ we need to decompose once more: Since $\Sigma w_* = h$, we find

$$\hat{h}_L = \Sigma_L^{1/2}\Big((\hat{h} - \hat{\Sigma}u_L) - (h - \Sigma u_L)\Big) + \Sigma_L^{1/2}(h - \Sigma u_L)$$
$$= \Sigma_L^{1/2}\Big((\hat{h} - \hat{\Sigma}u_L) - (h - \Sigma u_L)\Big) + \Sigma_L^{1/2}\Sigma R_L(\Sigma)w_* \ ,$$

satisfying

$$||\hat{h}_L|| \leq ||\Sigma_L^{1/2}\Big((\hat{h} - \hat{\Sigma}u_L) - (h - \Sigma u_L)\Big)|| + \sqrt{\mathcal{A}(L)} \ .$$

Applying Proposition 3 gives

$$||\hat{h}_L|| \leq \sqrt{\mathcal{A}(L)} + 12\log(8\delta^{-1})\left(\frac{\sqrt{\gamma L}(\kappa M + \kappa^2||u_L||)}{n} + \sqrt{\frac{\kappa^2 \gamma L \mathcal{A}(L) + \tilde{\sigma}^2\mathcal{N}(\frac{1}{\gamma L})}{n}}\right) \ , \quad (44)$$

with probability at least $1 - \delta/4$. Collecting (42), (43) and (44) yields

$$\hat{\mathcal{V}}(L) \leq C_{\tilde{\sigma},M,\kappa} \ \log^2(8\delta^{-1})\left(1 + \frac{\Delta(S,T)}{L^2}\right)^2$$
$$\left(\mathcal{A}(L) + \frac{\gamma L(1 + ||u_L||^2)}{n^2} + \frac{\mathcal{N}(1/\gamma L)}{n}\right) \ , \quad (45)$$

with probability at least $1 - \delta/4$, for some $C_{\tilde{\sigma},M,\kappa} < \infty$. Finally, Lemma 1 ensures that

$$||u_L|| = ||\Sigma G_L(\Sigma)w_*|| \leq ||w_*|| \ .$$

The bound in expectation follows from Lemma 11 by integration.

For the last part we refer to the proof of Lemma 2, from which we deduce that

$$1 + \frac{\Delta(S,T)}{L^2} \leq 1 + 2K \ ,$$

provided that $1 \leq K$ and $S \leq \frac{K-1}{K+1} T$. $\qquad\square$

### C.4 Main result on GD convergence

Proposition 1, Proposition 2 and Proposition 4 together lead our main result regarding the convergence of batch gradient descent, stated as the following theorem.

**Theorem 2.** *Let $1 \leq T$, $0 \leq S \leq T - 1$, Assumptions 1, 2 hold. Set $L = T - S$ and assume $\gamma\kappa^2 < 1$ as well as*

$$n \geq 16\kappa^2 \ \gamma L \ \max\{1, \mathcal{N}(1/(\gamma L))\} \ . \quad (46)$$

*1. If $0 \leq r \leq 1/2$ and $1 \leq K$, $0 \leq S \leq \frac{K-1}{K+1} T$, we have*

$$\mathbb{E}\Big[\ ||\Sigma^{1/2}(\bar{v}_L - w_*)||^2\ \Big] \leq C_r R^2 \ K^2 \ (\gamma L)^{-2(r+1/2)}$$
$$+ C_{\kappa,M,\sigma,\nu}K^2 \left(\mathcal{A}(L) + \frac{\gamma L(1 + ||w_*||)^2}{n^2} + \frac{\gamma L \ \mathcal{A}(L)}{n} + \frac{\mathcal{N}(\frac{1}{\gamma L})}{n}\right) \ ,$$

*for some $C_r < \infty$ and $C_{\kappa,M,\sigma,\nu} < \infty$.*

*2. If $1/2 < r$, $1 < K$, $0 < S \leq \frac{K-1}{K+1} T$ and $T \leq (K + 1)S$, we have*

$$\mathbb{E}\Big[\ ||\Sigma^{1/2}(\bar{v}_L - w_*)||^2\ \Big] \leq C_r \ C_K R^2 \left[(\gamma L)^{-2(r+1/2)} + \frac{1}{n}\right]$$
$$+ C_{\kappa,M,\sigma,\nu}C_K' \left(\mathcal{A}(L) + \frac{\gamma L(1 + ||w_*||)^2}{n^2} + \frac{\gamma L \ \mathcal{A}(L)}{n} + \frac{\mathcal{N}(\frac{1}{\gamma L})}{n}\right) \ ,$$

*for some $C_{\kappa,r} < \infty$ and $C_{\kappa,M,\sigma,\tilde{\tau}} < \infty$.*

From Theorem 2 we can immediately derive the Proof of Corollary 2.

**Corollary 2** (Rates of Convergence)**.** *Let any assumption of Theorem 2 hold and assume additionally Assumption 3. One has for any $n$ sufficiently large*

$$\mathbb{E}\Big[ \|\Sigma^{1/2}(\bar{v}_L - w_*)\|^2 \Big] \leq C\, n^{-\frac{2(r+1/2)}{2r+\nu}} \ ,$$

*under each of the following choices:*

1. *If $0 \leq r \leq 1/2$: $S = 0$, $\alpha, \beta \geq 0$ and*

$$\gamma_n \simeq n^{-\alpha} \quad T_n \simeq n^{\beta} \quad \text{such that} \quad \alpha - \beta = \frac{1}{2r+1+\nu} \ . \tag{47}$$

2. *If $1/2 < r$: $0 < S$, $S_n \asymp T_n$, with $T_n, \gamma_n$ as in (47).*

*Proof of Corollary 2.* Let $\gamma_n \simeq n^{-a}$, $L_n \simeq n^{\tilde{a}}$, with $a, \tilde{a} > 0$ satisfying $a - \tilde{a} = \frac{1}{2r+1+\nu}$. Plugging in Assumptions 2 and 3 gives in either case

$$\mathbb{E}\Big[ \|\Sigma^{1/2}(\bar{v}_{L_n} - w_*)\|^2 \Big] \leq C\Big( (\gamma_n L_n)^{-2(r+1/2)} + \frac{\gamma_n L_n}{n^2} + \frac{(\gamma_n L_n)^{-2r}}{n} + \frac{(\gamma_n L_n)^{\nu}}{n} + \frac{1}{n} \Big) \ ,$$

for some constant $C < \infty$, depending on all model parameters $\kappa, M, \nu, r, R$ and $\|w_*\|$. A short calculation shows that

$$n^{-1} = o\Big( (\gamma_n L_n)^{-2(r+1/2)} \Big) \ , \quad \frac{\gamma_n L_n}{n^2} = o\Big( (\gamma_n L_n)^{-2(r+1/2)} \Big)$$

and

$$\frac{(\gamma_n L_n)^{-2r}}{n} = o\Big( (\gamma_n L_n)^{-2(r+1/2)} \Big) \ ,$$

so we can disregard the terms $n^{-1}, \frac{\gamma_n L_n}{n^2}, \frac{(\gamma_n L_n)^{-2r}}{n}$ for $n$ large enough. The choice

$$\gamma_n L_n \simeq n^{\frac{1}{2r+1+\nu}}$$

precisely balances the two remaining terms $(\gamma_n L_n)^{-2(r+1/2)}$ and $\frac{(\gamma_n L_n)^{\nu}}{n}$. This choice also implies Assumption (46) if $n$ is sufficiently large. $\qquad\square$

# D  A general Result

Consider the recursion

$$\mu_{t+1} = \hat{Q}_{t+1}\, \mu_t + \gamma \xi_{t+1} \ , \qquad \hat{Q}_t = (I - \gamma \hat{H}_t) \ , \tag{48}$$

with $\mu_0 = 0$, with $\hat{H}_t$ linear i.i.d. random operators acting on $\mathcal{H}$ and with $\xi_t \in \mathcal{H}$ i.i.d. random variables, satisfying $\mathbb{E}[\xi_t] = 0$. For $0 \leq S \leq T - 1$ we let

$$\bar{\mu} := \bar{\mu}_{S,T} := \frac{1}{T-S} \sum_{t=S+1}^{T} \mu_t \ . \tag{49}$$

Denote $H = \mathbb{E}\Big[\hat{H}_t\Big]$. We assume that $Tr[H^{\alpha}] < \infty$ for some $\alpha \in (0,1]$ and

$$\mathbb{E}[\xi_t \otimes \xi_t] \preceq \sigma^2 H \ , \quad \mathbb{E}\Big[\hat{H}_t^2\Big] \preceq \kappa^2 H \ . \tag{50}$$

The last condition holds in particular when the $\hat{H}_t$ are bounded a.s. by $\kappa^2$. We generalize Proposition 1 given in [26] (see also [10]) to more general recursions and to tail-averaging, including full averaging and mini-batching as special cases.

**Proposition 5.** *Let $\alpha \in (0,1]$, $\gamma \kappa^2 \leq 1/4$ and $u \in [0, 1+\alpha]$. Under Assumption (50), one has*

$$\mathbb{E}\Big[ \|H^{\frac{u}{2}}\, \bar{\mu}_{S,T}\|^2 \Big] \leq 16\sigma^2\, Tr[H^{\alpha}] \gamma^{1-u+\alpha} (T-S)^{\alpha-u}\, \Upsilon(S,T) \ ,$$

*with $\Upsilon(S,T) = 1 + \frac{S+1}{T-S}$. If additionally $1 \leq K$ and $1 \leq T$, $0 \leq S \leq T-1$ satisfy $S \leq \frac{K-1}{K+1}\,T$, we have*

$$\Upsilon(S,T) \leq 1 + K \ .$$

The proof of this result is carried out in Section D.2. The basic idea is to derive a similar bound for the related *semi-stochastic recursion* (51), where $\hat{H}_t$ is replaced by it's expectation $H$, leaving the randomness in the noise variables $\xi_t$. This is done in Section D.1. In a second step one needs to control the difference between the full-stochastic recursion and the semi-stochastic iterates. This relies on a *perturbation argument*, summarized in Section D.2.

## D.1 Semi-Stochastic Recursion (SSR)

Let $H$ be a positive, self-adjoint operator on some Hilbert space $\mathcal{H}$, satisfying $H \preceq \kappa^2 I$. Consider the general recursion in $\mathcal{H}$

$$\mu_{t+1} = (1 - \gamma H)\mu_t + \gamma \xi_{t+1} \ , \tag{51}$$

with $\mu_0 = 0$ and $\gamma\kappa^2 < 1$. We further assume that

$$\mathbb{E}[\xi_t] = 0 \ , \qquad \mathbb{E}[\xi_t \otimes \xi_t] \preceq \sigma^2 H \ .$$

For $1 \leq T$ and $0 \leq S \leq T - 1$, we consider

$$\bar{\mu} := \bar{\mu}_{S,T} := \frac{1}{T - S} \sum_{t=S+1}^{T} \mu_t \ .$$

**Lemma 4** (SSR). *Let $\alpha \in (0, 1]$ and assume that $Tr(H^\alpha) < \infty$. Let $1 \leq T$. For any $u \in [0, 1 + \alpha]$ we have*

$$\mathbb{E}\Big[\|\, H^{u/2}\bar{\mu}\,\|^2\Big] \leq 4\sigma^2\, Tr[H^\alpha]\gamma^{1-u+\alpha}(T - S)^{\alpha-u}\, \Upsilon(S,T) \ ,$$

*with $\Upsilon(S,T) = 1 + \frac{S+1}{T-S}$. In particular, given $1 \leq K$ and if $0 \leq S \leq \frac{K-1}{K+1}T$ one has*

$$\Upsilon(S,T) \leq 1 + K \ .$$

*Proof of Lemma 4.* Setting $Q = 1 - \gamma H$, a standard calculation combined with the fact

$$\sum_{t=S+1}^{T} q^t = \frac{q^{S+1}(1 - q^{T-S})}{1 - q} \tag{52}$$

shows that the averaged iterates are given by

$$\bar{\mu} = \frac{\gamma}{T - S} \sum_{t=S+1}^{T} \sum_{k=0}^{t-1} Q^{t+1-k}\xi_k$$

$$= \frac{\gamma}{T - S} \sum_{t=0}^{S} \left(\sum_{k=S-t}^{T-(t+1)} Q^k\right)\xi_t + \frac{\gamma}{T - S} \sum_{t=S+1}^{T-1} \left(\sum_{k=0}^{T-(t+1)} Q^k\right)\xi_t$$

$$= \sum_{t=0}^{S} A_t\xi_t + \sum_{t=S+1}^{T-1} \tilde{A}_t\xi_t \ ,$$

where we set

$$A_t := \frac{\gamma}{T - S} \left(\sum_{k=S-t}^{T-(t+1)} Q^k\right) \ , \qquad \tilde{A}_t := \frac{\gamma}{T - S} \left(\sum_{k=0}^{T-(t+1)} Q^k\right) \ .$$

Thus, since $\mathbb{E}[\xi_t \otimes \xi_t] \preceq \sigma^2 H$, we find

$$\mathbb{E}\Big[\|\, H^{u/2}\bar{\mu}\,\|^2\Big] \leq 2\sum_{t=0}^{S} \mathbb{E}\big[Tr\big[H^u A_t^2\, \xi_t \otimes \xi_t\big]\big] + 2\sum_{t=S+1}^{T-1} \mathbb{E}\Big[Tr\Big[H^u \tilde{A}_t^2\, \xi_t \otimes \xi_t\Big]\Big]$$

$$= 2\sum_{t=0}^{S} Tr\big[H^u A_t^2\, \mathbb{E}[\xi_t \otimes \xi_t]\big] + 2\sum_{t=S+1}^{T-1} Tr\Big[H^u \tilde{A}_t^2\, \mathbb{E}[\xi_t \otimes \xi_t]\Big]$$

$$\leq \underbrace{2\sigma^2\sum_{t=0}^{S} Tr\big[H^{u+1} A_t^2\big]}_{\mathfrak{T}_1} + \underbrace{2\sigma^2\sum_{t=S+1}^{T-1} Tr\Big[H^{u+1} \tilde{A}_t^2\Big]}_{\mathfrak{T}_2} \ . \tag{53}$$

We proceed bounding the individual terms by applying (52). This gives

$$A_t = \frac{H^{-1}}{T-S} Q^{S-t} (1 - Q^{T-S}) \preceq \frac{H^{-1}}{T-S} (1 - Q^{T-S}) .$$

Furthermore,

$$
\begin{aligned}
Tr\big[H^{u-1}(1 - Q^{T-S})^2\big] &= \sum_{j \in \mathbb{N}} \sigma_j^{u-1}(1 - (1 - \gamma\sigma_j)^{T-S})^2 \\
&\leq \sum_{j \in \mathbb{N}} \sigma_j^{u-1}((T-S)\gamma\sigma_j)^{1-u+\alpha} \\
&= \gamma^{1-u+\alpha} \, Tr[H^\alpha] \, (T-S)^{1-u+\alpha} ,
\end{aligned}
$$

where in the inequality we use that for any $x \in [0,1]$, $u \in [0, 1+\alpha]$ one has

$$(1 - (1-x)^t)^2 \leq 1 - (1-x)^t \leq (tx)^{1-u+\alpha} .$$

As a result,

$$
\begin{aligned}
\mathcal{T}_1 &\leq 2\sigma^2 \frac{S+1}{(T-S)^2} \, Tr\big[H^{u-1}(1 - Q^{T-S})^2\big] \\
&\leq 2\sigma^2 \gamma^{1-u+\alpha} \, Tr[H^\alpha] \, (S+1) \, (T-S)^{\alpha-1-u} .
\end{aligned} \tag{54}
$$

Similarly,

$$\tilde{A}_t = \frac{H^{-1}}{T-S} (1 - Q^{T-t})$$

and

$$
\begin{aligned}
Tr\big[H^{u-1}(1 - Q^{T-t})^2\big] &= \sum_{j \in \mathbb{N}} \sigma_j^{u-1}(1 - (1 - \gamma\sigma_j)^{T-t})^2 \\
&\leq \gamma^{1-u+\alpha} \, Tr[H^\alpha] \, (T-t)^{1-u+\alpha} .
\end{aligned}
$$

Hence, since $1 - u + \alpha > 0$ we find

$$
\begin{aligned}
\mathcal{T}_2 &\leq \gamma^{1-u+\alpha} \frac{2\sigma^2}{(T-S)^2} \, Tr[H^\alpha] \sum_{t=S+1}^{T-1} (T-t)^{1-u+\alpha} \\
&= \gamma^{1-u+\alpha} \frac{2\sigma^2}{(T-S)^2} \, Tr[H^\alpha] \sum_{t=1}^{T-S-1} t^{1-u+\alpha} \\
&\leq \gamma^{1-u+\alpha} \frac{2\sigma^2}{(T-S)^2} \, Tr[H^\alpha](T-S-1)^{2-u+\alpha} \\
&\leq 2\sigma^2 \, \gamma^{1-u+\alpha} \, Tr[H^\alpha] \, (T-S)^{\alpha-u} .
\end{aligned} \tag{55}
$$

The result follows by combining (55), (54) and (53). $\qquad \square$

## D.2 Proof of Proposition 5

**Perturbation Argument.** Relating the semi-stochastic recursion (51) to the fully stochastic recursion in (48) is based on the perturbation idea from [1], which has been also applied in [10] and in [26] in a similar context. For sake of completeness we give a brief summary.

For $r \geq 0$ we introduce the sequence $(\mu_t^r)_t$

$$\mu_{t+1}^r = (I - H)\mu_t^r + \gamma \Xi_{t+1}^r ,$$

where $\Xi_t^0 = \xi_t$ and for $r \geq 0$

$$\Xi_{t+1}^{r+1} = (H - \hat{H}_t)\mu_t^r .$$

We further let $\eta_t^r = \mu_t - \sum_{j=0}^r \mu_t^j$ which follows the recursion

$$\eta_{t+1}^r = (I - \hat{H}_t)\eta_t^r + \gamma \Xi_{t+1}^{r+1} .$$

From Lemma 2 in [26][3] we have for any $r \geq 0$

$$\mathbb{E}[\mu_t^r \otimes \mu_t^r] \preceq \gamma^{r+1} \kappa^{2r} \sigma^2 I . \tag{56}$$

and

$$\mathbb{E}[\Xi_t^r \otimes \Xi_t^r] \preceq \gamma^r \kappa^{2r} \sigma^2 H . \tag{57}$$

Bounding $(\eta_t^r)_t$ is then done by applying the next Lemma, being an easy extension of Lemma 3 in [26] to tail-averaging.

**Lemma 5** (Rough Bound SGD Recursion). *Consider the SGD recursion given in* (48), *satisfying* (50). *Assume further* $\gamma \kappa^2 < 1$. *For any* $1 \leq T$, $0 \leq S \leq T - 1$ *we have*

$$\mathbb{E}\big[ \, ||H^{\frac{u}{2}} \, \bar{\mu}_{S,T}||^2 \, \big] \leq \sigma^2 \, \gamma^2 \kappa^u \, \frac{Tr[H]}{2(T-S)} \, \Delta(S,T) .$$

*where* $\Delta(S,T) = T(T+1) - S(S+1)$. *In particular, given* $1 \leq K$ *and if* $0 \leq S \leq \frac{K-1}{K+1} T$ *one has*

$$\Delta(S,T) = T(T+1) - S(S+1) \leq K(T-S)^2 .$$

*Proof of Lemma 5.* Following the arguments given in the proof of Lemma 3 in [26] we get

$$\mathbb{E}\big[ \, ||H^{\frac{u}{2}} \, \mu_t||^2 \, \big] \leq \sigma^2 \, \gamma^2 \kappa^u Tr[H] \, t .$$

By convexity, this leads to

$$\mathbb{E}\big[ \, ||H^{\frac{u}{2}} \, \bar{\mu}_{S,T}||^2 \, \big] \leq \frac{1}{T-S} \sum_{t=S+1}^{T} \mathbb{E}\big[||H^{\frac{u}{2}}\mu_t||^2\big]$$

$$\leq \sigma^2 \, \gamma^2 \kappa^u \frac{Tr[H]}{T-S} \sum_{t=S+1}^{T} t$$

$$= \sigma^2 \, \gamma^2 \kappa^u \frac{Tr[H]}{2(T-S)} (T(T+1) - S(S+1)) .$$

$\square$

**Proof of Proposition 5.** With these preparations we prove Proposition 5, applying the above described perturbation method. More precisely, we decompose

$$\bar{\mu}_{S,T} = \sum_{j=0}^{r} \bar{\mu}_{S,T}^j + \bar{\eta}_{S,T}^r$$

and have

$$\mathbb{E}\big[ \, ||H^{\frac{u}{2}} \, \bar{\mu}_{S,T}||^2 \, \big]^{1/2} \leq \sum_{j=0}^{r} \mathbb{E}\big[ \, ||H^{\frac{u}{2}} \, \bar{\mu}_{S,T}^j||^2 \, \big]^{1/2} + \mathbb{E}\big[ \, ||H^{\frac{u}{2}} \, \bar{\eta}_{S,T}^r||^2 \, \big]^{1/2} . \tag{58}$$

The first term in (58) we apply Lemma 4 and (57). Denoting

$$\Lambda(S,T) = 4\sigma^2 \, Tr[H^\alpha] \gamma^{1-u+\alpha} (T-S)^{\alpha-u} \left( 1 + \frac{S+1}{T-S} \right)$$

we get with $\gamma \kappa^2 \leq 1/4$

$$\sum_{j=0}^{r} \mathbb{E}\big[ \, ||H^{\frac{u}{2}} \, \bar{\mu}_{S,T}^j||^2 \, \big]^{1/2} \leq \sum_{j=0}^{r} \big(\gamma^j \kappa^{2j} \Lambda(S,T)\big)^{1/2}$$

$$= \sqrt{\Lambda(S,T)} \sum_{j=0}^{r} \big(\gamma \kappa^2\big)^{j/2}$$

$$\leq \frac{\sqrt{\Lambda(S,T)}}{1 - \sqrt{\gamma \kappa^2}}$$

$$\leq 2 \sqrt{\Lambda(S,T)} . \tag{59}$$

For bounding the second term in (58) we apply the rough SGD recursion bound from Lemma 5 and (57). Since $\gamma\kappa^2 < 1$, we find as $r \to \infty$

$$\mathbb{E}\big[ \, ||H^{\frac{u}{2}} \, \bar{\eta}^r_{S,T}||^2 \, \big]^{1/2} \leq \left( \gamma^{2+r} \kappa^{u+2r} \sigma^2 \frac{Tr[H]}{2(T-S)} \Delta(S,T) \right)^{1/2} \longrightarrow 0 \, . \tag{60}$$

The final result follows by combining (60) and (59) with (58).

# E  SGD Variance Term

Given $b \in [n]$ the mini-batch SGD recursion is given by

$$w_{t+1} = w_t + \gamma \frac{1}{b} \sum_{i=b(t-1)+1}^{bt} (\langle w_t, x_{j_i} \rangle_{\mathcal{H}} - y_{j_i}) x_{j_i} \, , \quad t = 1, ..., T \, ,$$

with $w_0 = 0$, $\gamma > 0$ a constant step-size[4] and where $j_1, ..., j_{bT}$ are i.i.d. random variables, distributed according to the uniform distribution on $[n]$.

We analyze tail-averaged mini-batch SGD. More precisely, for $0 \leq S \leq T - 1$ the algorithm under consideration is

$$\bar{w}_{S,T} := \frac{1}{T-S} \sum_{t=S+1}^{T} w_t \, .$$

For ease of notation we suppress dependence on $b$.

Recall the GD recursion

$$v_{t+1} = v_t - \gamma \frac{1}{n} \sum_{j=1}^{n} (\langle v_t, x_j \rangle_{\mathcal{H}} - y_j) x_j \, .$$

Denoting

$$\hat{\Sigma}_t = \frac{1}{b} \sum_{i=b(t-1)+1}^{bt} x_{j_i} \otimes x_{j_i} \, , \qquad \hat{h}_t = \frac{1}{b} \sum_{i=b(t-1)+1}^{bt} y_{j_i} x_{j_i}$$

for any $t \geq 1$, we have

$$w_{t+1} - v_{t+1} = \left(I - \gamma\hat{\Sigma}_{t+1}\right)(w_t - v_t) + \gamma\xi_{t+1} \, ,$$

where we define $\xi_{t+1} = \xi_{t+1}^{(1)} + \xi_{t+1}^{(2)}$ and

$$\xi_{t+1}^{(1)} = (\hat{\Sigma} - \hat{\Sigma}_{t+1})v_t \, , \quad \xi_{t+1}^{(2)} = \hat{h}_{t+1} - \hat{h} \, . \tag{61}$$

Denoting by $\mathcal{G}_n$ the $\sigma$- field generated by the data, we have for any $t \geq 1$

$$\mathbb{E}\Big[ \, \xi_{t+1}^{(1)} \mid \mathcal{F}_t, \mathcal{G}_n \, \Big] = \mathbb{E}\Big[ \, \xi_{t+1}^{(2)} \mid \mathcal{F}_t, \mathcal{G}_n \, \Big] = 0$$

almost surely. Thus, the difference $(\mu_t)_t = (w_t - v_t)_t$ follows a recursion as in (48), with $\hat{Q}_t = I - \gamma\hat{\Sigma}_{t+1}$.

**Proposition 6.** *Let* $\alpha \in (0,1]$, $\gamma\kappa^2 \leq 1/4$ *and* $n$ *be sufficiently large. Set* $L = T - S$.

$$\mathbb{E}\Big[ \, ||\Sigma^{\frac{1}{2}}(\bar{w}_{S,T} - \bar{v}_{S,T})||^2 \, \Big] \leq 32C_* \frac{\gamma^\alpha Tr[\Sigma^\alpha]}{bL^{1-\alpha}} \Upsilon(S,T) \, + \, 32\gamma^2\kappa^4 M^2 \frac{\tilde{\Delta}(S,T)^2}{L} \delta_n \, ,$$

*with* $C_* = \kappa^4(2||w_*|| + 1)^2 + M^2$,

$$\Upsilon(S,T) = 1 + \frac{S+1}{L} \, ,$$

$$\tilde{\Delta}(S,T) = \frac{1}{6}(T(T+1)(2T+1) - S(S+1)(2S+1))$$

*and*

$$\delta_n = 2\exp\left(-a\sqrt{\frac{n}{\gamma T \, \mathcal{N}(1/\gamma T)}}\right),$$

*for some $a > 0$. If additionally $1 \leq K$ and $1 \leq T$, $0 \leq S \leq T-1$ satisfy $S \leq \frac{K-1}{K+1} T$, we have*

$$\mathbb{E}\left[\,||\Sigma^{\frac{1}{2}}(\bar{w}_{S,T} - \bar{v}_{S,T})||^2\,\right] \leq 64 C_* K \frac{\gamma^\alpha Tr[\Sigma^\alpha]}{bL^{1-\alpha}} + 128\gamma^2 \kappa^4 M^2 \, K^4 L^5 \, \delta_n \,.$$

### E.1 Proof of Proposition 6

For proving Proposition 6 we aim at applying Proposition 5 and show that all assumptions are satisfied by stating a series of Lemmata. The first one provides an upper bound for the covariance of the noise process.

**Lemma 6.** *Assume $|Y| \leq M$ a.s. . For any $t = S, ..., T$ we have almost surely*

$$\mathbb{E}\left[\,\xi_{t+1}^{(1)} \otimes \xi_{t+1}^{(1)} \mid \mathcal{G}_n\,\right] \preceq \frac{\kappa^4}{b} \, ||v_t||^2 \, \hat{\Sigma}$$

*and*

$$\mathbb{E}\left[\,\xi_{t+1}^{(2)} \otimes \xi_{t+1}^{(2)} \mid \mathcal{G}_n\,\right] \preceq \frac{M^2}{b} \, \hat{\Sigma} \,.$$

*Here, expectation is taken with respect to the b- fold uniform distribution on $[n]$ in step $t+1$.*

*Proof of Lemma 6.* Recall that

$$\xi_{t+1}^{(1)} = (\hat{\Sigma} - \hat{\Sigma}_{t+1})v_t = \frac{1}{b} \sum_{i=bt+1}^{b(t+1)} \tilde{\xi}_i \,,$$

with

$$\tilde{\xi}_i := \hat{\Sigma}v_t - \langle v_t, x_{j_i}\rangle x_{j_i} \,.$$

By independence, we have

$$\mathbb{E}\left[\,\xi_{t+1}^{(1)} \otimes \xi_{t+1}^{(1)} \mid \mathcal{G}_n\,\right] = \frac{1}{b^2} \sum_{i,i'} \mathbb{E}\left[\,\tilde{\xi}_i \otimes \tilde{\xi}_{i'} \mid \mathcal{G}_n\,\right]$$

$$= \frac{1}{b^2} \sum_i \mathbb{E}\left[\,\tilde{\xi}_i \otimes \tilde{\xi}_i \mid \mathcal{G}_n\,\right] \,.$$

The first part follows then by

$$\mathbb{E}\left[\,\tilde{\xi}_i \otimes \tilde{\xi}_i \mid \mathcal{G}_n\,\right] \preceq \mathbb{E}\left[\,\langle v_t, x_{j_i}\rangle_{\mathcal{H}}^2 \, x_{j_i} \otimes x_{j_i} \mid \mathcal{G}_n\,\right] \preceq \kappa^4 \, ||v_t||^2 \, \hat{\Sigma} \,.$$

The second part of the Lemma follows by writing

$$\xi_{t+1}^{(2)} = \hat{h}_{t+1} - \hat{h} = \frac{1}{b} \sum_{i=b(t-1)+1}^{bt} \xi'_{j_i} \,,$$

with $\xi'_{j_i} = y_{j_i} x_{j_i} - \hat{h}$ and observing that

$$\mathbb{E}\left[\,\xi'_{j_i} \otimes \xi'_{j_i} \mid \mathcal{G}_n\,\right] \preceq \mathbb{E}\left[\,|y_{j_i}|^2 \, x_{j_i} \otimes x_{j_i} \mid \mathcal{G}_n\,\right] \preceq M^2 \, \hat{\Sigma} \,. \tag{62}$$

$\qquad\qquad\qquad\qquad\qquad\qquad\qquad\qquad\qquad\qquad\qquad\qquad\qquad\qquad\qquad\qquad\qquad\qquad\qquad\square$

The next Lemma provides a uniform for the GD updates, leading to a uniform bound for the noise process.

**Lemma 7** (Uniform Bound Gradient Descent updates). *Assume $|Y| \leq M$ a.s. and let $\tilde{M} = \max(M, \kappa||w_*||)$ and $\bar{\sigma} := 2\tilde{M}$. For any $\delta \in (0,1]$ and for any $S+1 \leq t \leq T$, with probability at least $1 - \delta$ one has*

$$||v_t|| \leq 2\,||w_*|| + 1\,,$$

*provided*

$$n \geq 64 \max\{\bar{\sigma}^2, \kappa\tilde{M}\} \log^2(2\delta^{-1})\gamma T \max\{1, \mathcal{N}(1/\gamma T)\}\,.$$

*Moreover, with probability at least $1 - (T-S)\delta$ one has*

$$\sup_{S+1 \leq t \leq T} ||v_t|| \leq 2\,||w_*|| + 1\,. \tag{63}$$

*Proof of Lemma 7.* We decompose

$$||v_t|| \leq ||v_t - w_*|| + ||w_*||\,.$$

For bounding the first term we apply the results in [5], decomposition (5.9) with eq. (5.17) and (5.22) for $\lambda = \frac{1}{\gamma t}$[5]. For that we need to ensure a moment condition

$$\mathbb{E}\big[|Y - \langle w_*, X\rangle|^l \,|X\big] \leq \frac{1}{2}l!\bar{\sigma}^2\tilde{M}^{l-2} \quad \text{a.s.}\,, \tag{64}$$

for some $\bar{\sigma}^2 > 0$, $\tilde{M} < \infty$ and for any $l \geq 2$. Indeed, since $|Y| \leq M$ a.s. and $|\langle w_*, X\rangle| \leq \kappa||w_*||$, we easily derive

$$\begin{aligned}
\mathbb{E}\big[|Y - \langle w_*, X\rangle|^l \,|X\big] &\leq 2^{l-1}\big(\mathbb{E}\big[|Y|^l|X\big] + |\langle w_*, X\rangle|^l\big) \\
&\leq 2^{l-1}\big(M^l + (\kappa||w_*||)^l\big) \\
&\leq \frac{1}{2}l!\bar{\sigma}^2\tilde{M}^{l-2} \quad \text{a.s.}\,,
\end{aligned}$$

with $\tilde{M} = \max(M, \kappa||w_*||)$ and $\bar{\sigma} := 2\tilde{M}$. Thus, with probability at least $1 - \delta$

$$\begin{aligned}
||v_t - w_*|| &\leq ||w_*|| + 2\log(2\delta^{-1})\left(\frac{\kappa\tilde{M}\gamma t}{n} + \bar{\sigma}\sqrt{\frac{\gamma t\,\mathcal{N}(1/\gamma t)}{n}}\right) \\
&\leq ||w_*|| + 2\log(2\delta^{-1})\left(\frac{\kappa\tilde{M}\gamma T}{n} + \bar{\sigma}\sqrt{\frac{\gamma T\,\mathcal{N}(1/\gamma T)}{n}}\right)\,.
\end{aligned}$$

Assuming

$$n \geq 64 \max\{\bar{\sigma}^2, \kappa\tilde{M}\} \log^2(2\delta^{-1})\gamma T \max\{1, \mathcal{N}(1/\gamma T)\} \tag{65}$$

we find

$$2\log(2\delta^{-1})\bar{\sigma}\sqrt{\frac{\gamma T\,\mathcal{N}(1/\gamma T)}{n}} \leq \frac{1}{4}\,.$$

Moreover, the same condition also implies

$$n \geq 64\kappa\tilde{M}\log(2\delta^{-1})\gamma T$$

owing to the fact that $2\log(2\delta^{-1}) > 1$ and thus

$$2\log(2\delta^{-1})\frac{\kappa\tilde{M}\gamma T}{n} \leq \frac{1}{32}\,.$$

Hence,

$$||v_t|| \leq 2||w_*|| + \frac{1}{32} + \frac{1}{4} \leq 2||w_*|| + 1\,,$$

with probability at least $1 - \delta$. The uniform bound in (63) follows from taking a union bound, i.e.

$$\left\{\sup_{S+1 \leq t \leq T} ||v_t|| \geq 2||w_*|| + 1\right\} \subseteq \bigcup_{t=S+1}^{T}\{||v_t|| \geq 2||w_*|| + 1\}\,.$$

$\square$

**Lemma 8** ([6], eq. (47)). *For any $\delta \in (0, 1]$ and $\lambda > 0$ satisfying*

$$n\lambda \geq 64\kappa^2 \log^2(2\delta^{-1}) \max\{1, \mathcal{N}(\lambda)\} \tag{66}$$

*one has*

$$\left\| \left(\hat{\Sigma} + \lambda\right)^{-1}(\Sigma + \lambda) \right\| \leq 2$$

*with probability at least $1 - \delta$.*

The following Lemma provides a rough bound for the tail-averaged updates, generalized from [26] to tail-averaging.

**Lemma 9** (Rough bound for averaged SGD variance). *Assume $|Y| \leq M$ a.s. and $\gamma\kappa^2 < 1$. One has almost surely*

$$\|\bar{w}_{S,T} - \bar{v}_{S,T}\| \leq 4\gamma\kappa M \frac{\tilde{\Delta}(S,T)}{T-S} ,$$

*where*

$$\tilde{\Delta}(S,T) = \sum_{t=S+1}^{T} t^2 = \frac{1}{6}(T(T+1)(2T+1) - S(S+1)(2S+1)) .$$

*Moreover, if $1 \leq K$, $1 \leq T$, $0 \leq S \leq T - 1$ satisfy $S \leq \frac{K-1}{K+1} T$, one has*

$$\tilde{\Delta}(S,T) \leq 2K^2(T-S)^3$$

*and*

$$\|\bar{w}_{S,T} - \bar{v}_{S,T}\| \leq 8\gamma\kappa M \; K^2 \; (T-S)^2 ,$$

*almost surely.*

*Proof of Lemma 9.* Recall that the gradient updates are given by $v_0 = 0$ and

$$v_{t+1} = v_t - \gamma(\hat{\Sigma}v_t - \hat{h}) = \hat{Q}v_t + \gamma\hat{h} ,$$

with $\hat{Q} = (1 - \gamma\hat{\Sigma})$, $\|\hat{Q}\| < 1$ and $\|\hat{h}\| \leq \kappa M$. Thus,

$$\|v_{t+1}\| \leq \|v_t\| + \gamma\kappa M$$

and inductively one obtains

$$\|v_t\| \leq \gamma\kappa M \; t . \tag{67}$$

Let $\mu_t = w_t - v_t$. Starting with $\mu_0 = 0$, then $(\mu_t)_t$ follows the recursion

$$\mu_{t+1} = \hat{Q}_{t+1}\mu_t + \gamma\xi_{t+1}, \qquad \hat{Q}_{t+1} = (I - \gamma\hat{\Sigma}_{t+1}) ,$$

where $\xi_{t+1} = \xi_{t+1}^{(1)} + \xi_{t+1}^{(2)}$ is defined in (22). By (67) and since $\gamma\kappa^2 < 1$ we have

$$\|\xi_{t+1}^{(1)}\| \leq \|(\hat{\Sigma} - \hat{\Sigma}_{t+1})\| \; \|v_t\| \leq 2\gamma\kappa^3 M \; t < 2\kappa M \; t .$$

Furthermore,

$$\|\xi_{t+1}^{(2)}\| = \|\hat{h}_{t+1} - \hat{h}\| \leq 2\kappa M .$$

Using $\|\hat{Q}_{t+1}\| < 1$, one easily calculates

$$\|\mu_t\| \leq \gamma \sum_{j=1}^{t} \|\xi_j\| \leq 4\gamma\kappa M \; t^2 .$$

Thus,

$$\|\bar{\mu}_{S,T}\| \leq \frac{4\gamma\kappa M}{T-S} \sum_{t=S+1}^{T} t^2 = 4\gamma\kappa M \frac{\tilde{\Delta}(S,T)}{T-S} ,$$

with

$$\tilde{\Delta}(S,T) = \sum_{t=S+1}^{T} t^2 = \frac{1}{6}(T(T+1)(2T+1) - S(S+1)(2S+1)) .$$

Finally,

$$\tilde{\Delta}(S,T) \leq 2K^2(T-S)^3 ,$$

implied by $S \leq T - 1$ and $S \leq \frac{K-1}{K+1} T$. $\qquad\square$

*Proof of Proposition 6.* We define the events

$$\mathcal{E}_1 = \left\{ \mathbf{x} \in \mathcal{X}^n \ : \ \left\| \left( \hat{\Sigma} + \lambda \right)^{-1/2} (\Sigma + \lambda)^{1/2} \right\|^2 \leq 2 \right\}, \tag{68}$$

where we set $\lambda = \frac{1}{\gamma L}$ and

$$\mathcal{E}_2 = \left\{ (\mathbf{x}, \mathbf{y}) \in \mathcal{X}^n \times \mathcal{Y}^n \ : \ \forall t = S+1, ..., T \ : \ \mathbb{E}[\ \xi_{t+1} \otimes \xi_{t+1} \mid \mathcal{G}_n\ ] \preceq \frac{C_*}{b} \hat{\Sigma} \right\}, \tag{69}$$

with $C_* = \kappa^4 (2\|w_*\| + 1)^2 + M^2$.
Denoting $\bar{\mathcal{N}}(\lambda) = \max\{1, \mathcal{N}(\lambda)\}$, Lemma 8 gives $\mathbb{P}[\mathcal{E}_1^c] \leq \delta_1$, provided

$$n \geq 64\kappa^2 \log^2(2\delta_1^{-1}) \gamma L \bar{\mathcal{N}}(1/\gamma L)$$

or, equivalently,

$$\delta_1 \geq 2 \exp\left( -a_1 \sqrt{\frac{n}{\gamma L \bar{\mathcal{N}}(1/\gamma L)}} \right), \tag{70}$$

with $a_1 = \frac{1}{8\kappa}$. Similarly, applying Lemma 7 and Lemma 6 gives $\mathbb{P}[\mathcal{E}_2^c] \leq L\delta_2$ if

$$n \geq C_{\kappa,\tilde{M},\bar{\sigma}} \log^2(2\delta_2^{-1}) \gamma T \bar{\mathcal{N}}(1/\gamma T), \quad C_{\kappa,\tilde{M},\bar{\sigma}} = 64 \max\{\bar{\sigma}^2, \kappa\tilde{M}\}$$

or equivalently

$$\delta_2 \geq 2 \exp\left( -a_2 \sqrt{\frac{n}{\gamma T \bar{\mathcal{N}}(1/\gamma T)}} \right). \tag{71}$$

with $a_2 = \frac{1}{\sqrt{C_{\kappa,\tilde{M},\bar{\sigma}}}}$.
Setting $\bar{\mu}_{S,T} = \bar{w}_{S,T} - \bar{v}_{S,T}$, we decompose

$$\mathbb{E}\left[ \|\Sigma^{\frac{1}{2}} \bar{\mu}_{S,T}\|^2 \right] \leq \mathbb{E}\left[ \|\Sigma^{\frac{1}{2}} \bar{\mu}_{S,T}\|^2 \ 1_{\mathcal{E}_1 \cap \mathcal{E}_2} \right] + \mathbb{E}\left[ \|\Sigma^{\frac{1}{2}} \bar{\mu}_{S,T}\|^2 \ 1_{\mathcal{E}_1^c} \right] + \mathbb{E}\left[ \|\Sigma^{\frac{1}{2}} \bar{\mu}_{S,T}\|^2 \ 1_{\mathcal{E}_2^c} \right]. \tag{72}$$

For bounding the first term note that

$$\Sigma^{\frac{1}{2}} = \Sigma^{\frac{1}{2}} (\Sigma + \lambda)^{-\frac{1}{2}} (\Sigma + \lambda)^{\frac{1}{2}} (\hat{\Sigma} + \lambda)^{-\frac{1}{2}} (\hat{\Sigma} + \lambda)^{\frac{1}{2}},$$

where

$$\|\Sigma^{\frac{1}{2}} (\Sigma + \lambda)^{-\frac{1}{2}}\| \leq 1.$$

Thus, by definition of $\mathcal{E}_1$ and $\mathcal{E}_2$, using $\|(\hat{\Sigma}+\lambda)^{\frac{1}{2}} u\|^2 = \|\hat{\Sigma}^{\frac{1}{2}} u\|^2 + \lambda\|u\|^2$, we find with $\lambda = \frac{1}{\gamma(T-S)}$ and Proposition 5 with $\sigma^2 = C_*/b$

$$\begin{aligned} \mathbb{E}\left[ \|\Sigma^{\frac{1}{2}} \bar{\mu}_{S,T}\|^2 \ 1_{\mathcal{E}_1 \cap \mathcal{E}_2} \right] &\leq 2\mathbb{E}\left[ \|\hat{\Sigma}^{\frac{1}{2}} \bar{\mu}_{S,T}\|^2 \right] + \frac{2}{\gamma(T-S)} \mathbb{E}\left[ \|\bar{\mu}_{S,T}\|^2 \right] \\ &\leq 32 C_* \frac{\gamma^\alpha \Upsilon(S,T)}{b L^{1-\alpha}} \mathbb{E}\left[ Tr\left[ \hat{\Sigma}^\alpha \right] \right] \\ &\leq 32 C_* \frac{\gamma^\alpha \Upsilon(S,T)}{b L^{1-\alpha}} Tr[\Sigma^\alpha]. \end{aligned} \tag{73}$$

In the last step we apply Jensen's inequality, giving $\mathbb{E}\left[ Tr\left[ \hat{\Sigma}^\alpha \right] \right] \leq Tr[\Sigma^\alpha]$.

For bounding the second and third term recall that $\|\Sigma^{\frac{1}{2}}\|^2 \leq \kappa^2$. We have by Lemma 9

$$\|\Sigma^{\frac{1}{2}} \bar{\mu}_{S,T}\|^2 \leq 16\gamma^2 \kappa^4 M^2 \frac{\tilde{\Delta}(S,T)^2}{L^2}.$$

Hence,

$$\mathbb{E}\left[ \|\Sigma^{\frac{1}{2}} \bar{\mu}_{S,T}\|^2 \ 1_{\mathcal{E}_1^c} \right] \leq 16\gamma^2 \kappa^4 M^2 \frac{\tilde{\Delta}(S,T)^2}{L^2} \delta_1 \tag{74}$$

and

$$\mathbb{E}\left[ \|\Sigma^{\frac{1}{2}} \bar{\mu}_{S,T}\|^2 \ 1_{\mathcal{E}_2^c} \right] \leq 16\gamma^2 \kappa^4 M^2 \frac{\tilde{\Delta}(S,T)^2}{L^2} L \delta_2. \tag{75}$$

The result follows from collecting (75), (74), (73) and (72) and by choosing

$$\delta_n := \max\{\delta_1, \delta_2\} = 2\exp\left(-a\sqrt{\frac{n}{\gamma T\ \bar{N}(1/\gamma T)}}\right) , \tag{76}$$

with $a = \min\{a_1, a_2\}$. Note that we also use the fact that $\gamma t\bar{N}(1/\gamma t)$ is increasing in $t$ and $L \leq T$.

If additionally $1 \leq K$ and $1 \leq T$, $0 \leq S \leq T-1$ satisfy $S \leq \frac{K-1}{K+1} T$, we have

$$\Upsilon(S,T) \leq 1 + K \leq 2K , \qquad \tilde{\Delta}(S,T) \leq 2K^2 L^3 .$$

this gives

$$\mathbb{E}\left[||\Sigma^{\frac{1}{2}}\bar{\mu}_{S,T}||^2\right] \leq 64C_*K\ \frac{\gamma^\alpha Tr[\Sigma^\alpha]}{bL^{1-\alpha}} + 128\gamma^2\kappa^4 M^2\ K^4 L^5\ \delta_n .$$

$\square$

# F   Main Results Tail-Averaging SGD

From Theorem 2 and Proposition 6 combined with decomposition (28) we obtain

**Theorem 3.** *Let $\alpha \in (0,1]$, $1 \leq T$, $0 \leq S \leq T-1$ and Assumptions 1, 2 hold. Assume $\gamma\kappa^2 < 1/4$. Set $L = T - S$ and*

$$\delta_n = 2\exp\left(-a\sqrt{\frac{n}{\gamma T\ N(1/\gamma T)}}\right) ,$$

*with $a > 0$ given in (76). Then*

$$\mathbb{E}\left[\ ||\Sigma^{\frac{1}{2}}(\bar{w}_{S,T} - w_*)||^2\ \right] \lesssim \frac{\gamma^\alpha}{bL^{1-\alpha}}\ Tr[\Sigma^\alpha] + (\gamma L)^{-2(r+1/2)} + \frac{N(\frac{1}{\gamma L})}{n}$$
$$+ \frac{\gamma L}{n^2} + \frac{(\gamma L)^{-2r}}{n} + \frac{1}{n} + \gamma^2 L^5\ \delta_n ,$$

*under each of the following assumptions:*

1. *$0 \leq r \leq 1/2$ and $1 \leq K$, $0 \leq S \leq \frac{K-1}{K+1} T$,*

2. *$1/2 < r$, $1 < K$, $0 < S \leq \frac{K-1}{K+1} T$ and $T \leq (K+1)S$.*

*The constant hidden in $\lesssim$ in the above bound depends on the model parameters $\kappa, M, r, R, K$ given in the assumptions.*

*Proof of Corollary 1.*  Plugging in Assumptions 2 and 3 gives in either case

$$\mathbb{E}\left[\ ||\Sigma^{1/2}(\bar{w}_{L_n} - w_*)||^2\ \right] \lesssim \frac{\gamma_n^\alpha}{b_n L_n^{1-\alpha}} + (\gamma_n L_n)^{-2(r+1/2)} + \frac{(\gamma_n L_n)^\nu}{n}$$
$$+ \frac{\gamma_n L_n}{n^2} + \frac{(\gamma_n L_n)^{-2r}}{n} + \frac{1}{n} + \gamma_n^2 L_n^5\ \delta_n ,$$

As in the proof of Corollary 2 we have as $n \to \infty$

$$n^{-1} = o\left((\gamma_n L_n)^{-2(r+1/2)}\right) , \qquad \frac{\gamma_n L_n}{n^2} = o\left((\gamma_n L_n)^{-2(r+1/2)}\right)$$

and

$$\frac{(\gamma_n L_n)^{-2r}}{n} = o\left((\gamma_n L_n)^{-2(r+1/2)}\right) ,$$

so we can disregard the terms $n^{-1}, \frac{\gamma_n L_n}{n^2}, \frac{(\gamma_n L_n)^{-2r}}{n}$ for $n$ large enough. Furthermore, $\delta_n$ satisfies

$$\delta_n \lesssim \exp\left(-a\sqrt{\frac{n}{(\gamma_n T_n)^{\nu+1}}}\right) = \exp\left(-a\ n^{\frac{1}{2}(1-\frac{\nu+1}{2r+1+\nu})}\right) ,$$

showing

$$\gamma_n^2 L_n^5 \, \delta_n = o\Big((\gamma_n L_n)^{-2(r+1/2)}\Big)$$

as $n \to \infty$ since $1 - \frac{\nu+1}{2r+1+\nu} > 0$ and $\delta_n$ decreases exponentially fast (note that we require $S_n$ to be of the same order as $T_n$). Furthermore, the choice

$$\gamma_n L_n \simeq n^{\frac{1}{2r+1+\nu}}$$

precisely balances the two terms $(\gamma_n L_n)^{-2(r+1/2)}$ and $\frac{(\gamma_n L_n)^\nu}{n}$, so the remaining leading order terms are

$$\mathbb{E}\Big[\, ||\Sigma^{1/2}(\bar{w}_{L_n} - w_*)||^2 \,\Big] \lesssim \frac{\gamma_n^\alpha}{b_n L_n^{1-\alpha}} + (\gamma_n L_n)^{-2(r+1/2)} \; .$$

Finally, choosing $\alpha = \nu$, a calculation shows that all choices of $b_n, (\gamma_n L_n)$ are balancing the two remaining terms. $\qquad\square$

## G   Auxiliary Technical Lemmata

### G.1   Probabilistic Ones

**Proposition 7** ([14], Proposition 1). *Define*

$$\mathcal{B}_n(\lambda) := \left[1 + 4\kappa^2\left(\frac{\kappa}{n\lambda} + \sqrt{\frac{\mathcal{N}(\lambda)}{n\lambda}}\right)^2\right]. \tag{77}$$

*For any $\lambda > 0$, $\delta \in (0,1]$, with probability at least $1 - \delta$ one has*

$$\left\|(\hat{\Sigma} + \lambda)^{-1}(\Sigma + \lambda)\right\| \le 8\log^2(2\delta^{-1})\mathcal{B}_n(\lambda) \; . \tag{78}$$

**Corollary 3.** *Let $\delta \in (0,1]$ and assume that*

$$n\lambda \ge 16\kappa^2 \max\{1, \mathcal{N}(\lambda)\} \; . \tag{79}$$

*Then*

$$\mathcal{B}_n(\lambda) \le 2 \; .$$

*In particular,*

$$\left\|\left(\hat{\Sigma} + \lambda\right)^{-1}(\Sigma + \lambda)\right\| \le 16\log^2(2\delta^{-1})$$

*holds with probability at least $1 - \delta$.*

*Proof of Corollary 3.* Assumption (79) immediately gives

$$\sqrt{\frac{\mathcal{N}(\lambda)}{n\lambda}} \le \frac{1}{4\kappa}$$

as well as

$$\frac{\kappa}{\lambda n} \le \frac{1}{4\kappa} \; .$$

The result then follows by plugging these bounds into (77). $\qquad\square$

**Lemma 10.** *Let $\lambda > 0$ and assume that*

$$n\lambda \ge 16\kappa^2 \max\{1, \mathcal{N}(\lambda)\}. \tag{80}$$

*For any $w \in \mathcal{H}$ and $\delta \in (0,1]$, one has with probability at least $1 - \delta$*

$$||\Sigma^{\frac{1}{2}}w|| \le 4\log(2\delta^{-1}) \, ||(\hat{\Sigma} + \lambda)^{\frac{1}{2}}w|| \; .$$

*Proof of Lemma 10.* Applying Corollary 3, we find

$$||\Sigma^{\frac{1}{2}}w|| \le ||\Sigma^{\frac{1}{2}}(\Sigma+\lambda)^{-\frac{1}{2}}|| \; ||(\hat{\Sigma}+\lambda)^{-\frac{1}{2}}(\Sigma+\lambda)^{\frac{1}{2}}|| \; ||(\hat{\Sigma}+\lambda)^{\frac{1}{2}}w||$$
$$\le 4\log(2\delta^{-1}) \, ||(\hat{\Sigma}+\lambda)^{\frac{1}{2}}w|| \;.$$

$\square$

**Lemma 11.** *Let $X$ be a nonnegative random variable with $\mathbb{P}[X > C\log^u(k\delta^{-1})] < \delta$ for any $\delta \in (0,1]$. Then $\mathbb{E}[X] \le \frac{C}{k}u\Gamma(u)$, where $\Gamma$ denotes the Gamma-function.*

*Proof.* Apply $\mathbb{E}[X] = \int_0^\infty \mathbb{P}[X > t]dt$. $\square$

## G.2 Miscellaneous

**Lemma 12.** *For any $0 \le S \le T$ and for any $a \in [0,1]$ one has*

$$T^{a+1} - S^{a+1} \le (T+S)(T-S)^a \;. \tag{81}$$

*Proof of Lemma 12.* Rewriting (81) to

$$1 - \left(\frac{S}{T}\right)^{a+1} \le \left(1+\frac{S}{T}\right)\left(1-\frac{S}{T}\right)^a$$

shows that it is sufficient to show that

$$h_a(u) := (1+u)(1-u)^a + u^{a+1} - 1 \ge 0$$

for any $u \in [0,1]$. This follows by observing that $h_0(u) \equiv 0$, $h_1(u) = 2u$. Moreover, $h_a$ is concave if $a \in (0,1)$, satisfying $h_a(0) = h_a(1) = 0$. $\square$

**Lemma 13.** *Let $(a_k)_k$ and $(\xi_k)_k$ be two sequences, then*

$$\sum_{t=S+1}^{T}\sum_{k=0}^{t-1} a_{t-1-k}\,\xi_k = \sum_{t=0}^{S}\left(\sum_{k=S-t}^{T-(t+1)} a_k\right)\xi_t + \sum_{t=S+1}^{T-1}\left(\sum_{k=0}^{T-(t+1)} a_k\right)\xi_t \;.$$

**Lemma 14.** *1. Let $\varphi : \mathbb{R}_+ \longrightarrow \mathbb{R}_+$ monotonically non-decreasing. Then*

$$\sum_{t=S}^{T}\varphi(t) \;\le\; \int_S^{T+1}\varphi(t)\,dt \;\le\; \sum_{t=S}^{T}\varphi(t+1) \;.$$

*2. Let $\varphi : \mathbb{R}_+ \longrightarrow \mathbb{R}_+$ monotonically non-increasing. Then*

$$\sum_{t=S}^{T}\varphi(t+1) \;\le\; \int_S^{T+1}\varphi(t)\,dt \;\le\; \sum_{t=S}^{T}\varphi(t) \;.$$

## Footnotes

[3]Lemma 2 in [26] is shown in the special case where $\hat{H}_t = z_t \otimes z_t$ for i.i.d. observations $z_t \in \mathcal{H}$, but the proof of (56) and (57) is literally the same.

[4] *constant* means independent of the iteration $t$, but possibly depending on $n$

[5]The constant in eq. (5.17) equals one in case of GD.