[Reviews · NeurIPS 2019]

Reviewer 1



*********After author response********** I thank the authors for their response. The response addressed the clarity issues I raised. While I understand the high-level intuition that tail-averaging allows larger step sizes by reducing the variance, the authors could do a better job by making this intuition clear in the paper and supporting it by a quantitative relationship between tail-length and step-size. Therefore, I still keep my evaluation. ********************************************** Originality: While there are many previous analyses of least-squares learning in Hilbert spaces, the finding that tail-averaging overcomes saturation is novel and interesting, providing deeper understanding of the role of averaging in SGD. Quality: I didn't go through the appendices, but the analysis and proofs in the main paper are well-supported and sound. The experiments support the theory well. Clarity: The paper is clearly written in general and Section 3 provided a nice intuition, but I find several small spots that confused me (see the entries 2-6 in the "improvements" part). Significance: The paper seems to suggest the use of tail-averaging instead of full-averaging in practice, but the reason for using averaging in the first place is to allow larger step sizes. The paper seems not clear about why tail-averaging allows larger step sizes than no averaging, making the result less significant.

Reviewer 2



*************After author response************ Thank you for the answer. I'll keep my mark and vote for accepting this paper. **************************************************** In this paper, the authors provides a clear analysis for Stochastic Gradient Descent for least-square in a non-parametric setting on the interplay of: -the number of passes over the data set -the size of the mini-batches -the size of tail-averages -the step-size of the stochastic gradient descent Originality: As far as I understand the paper, it does not seem that the analysis of each term of the SGD estimator uses now tools or really new technique, even if the analysis of convergence of the population risk gradient descent iterates to the real minimizer considered with tail-averaging is new. Yet, the techniques for bounding each term seem borrowed and adapted from previous papers analyzing SGD for least-squares problems -related papers are adequately cited. Quality and clarity : Theoretically speaking, the paper is self-contained and provides proofs of all theorems and a clear discussion on all the assumptions made in the paper. Furthermore, despite the number of parameters concerned with the analysis, the main results (Theorem 1 and Corollary 1) are very clear and clearly compared with the relative work. However, the experimental section may lack of a real dataset where r can be computed and where we could see the difference between tail and uniform averaging. Moreover, when Theorem 1 and Corollary 1 speak about the interplay between 4 different parameters, the experimental section only provides a clear illustration of the difference between tail and uniform averaging (even if this is the main result of the paper). Significance: The paper is very clear, very well-written, and is a clear summary for SGD for Least-Square in the non-parametric setting but lack from real novel ideas.

Reviewer 3



I am not an expert of this area. I some questions for you: have you evaluated your methods on real scenario instead of only on synthetic data? Does the conclusion still hold on other objective functions such as cross entropy?

[Author Response · NeurIPS 2019]

We thank all reviewers for reading our paper and appreciating our results. We respond to each reviewer's comments
below.

**Response for Reviewer #2.**

1. In a classical stochastic approximation framework it is well known that choosing too small step sizes can
lead to slow convergence while larger step sizes improve convergence but suppress noise poorly [see e.g.
Nemirovski et al. 2009]. In order to suppress, to some extent, noisy trajectories while keeping large step sizes
and thus fast convergence, a common approach is to take appropriate averages of the iterates. The intuition
behind tail-averaging is that an appropriate choice of the tail-length realizes a trade-off between robustness
(less noise) and fast convergence. In a nutshell, averaging allows larger step-size since it reduces the variance
of SGD. Tail averaging with sufficiently "long" tail preserve this benefit. Characterizing the exact length which
allows this effect is one of the contribution of our analysis.

2. By the saturation effect and the improvements provided by tail-averaging being "purely deterministic", we
mean that both already manifest themselves when considering (deterministic) population gradient descent
rather than stochastic gradient descent. Specifically, we show in Section 3 that population gradient descent
with full averaging suffers from a saturation effect, which can be remedied by tail-averaging. Our detailed
analysis of SGD shows that the same observation can be used to obtain improved bounds on the approximation
error.

3. In line 219, we refer to the contributions of Jain et al. [19], listed in their Section 3. We will rephrase this
sentence to avoid potential confusion in the final version.

4. Correct, thanks for pointing this out!

5. Also correct, thanks!

6. Indeed, the first experiment shows the results after a single pass over the data. We will make this clear in the
final version.

**Response for Reviewer #4.**     You are correct to point out that the bulk of our analysis uses tools that have been known
before, but nevertheless the observation that tail-averaging can eliminate the saturation effect associated with uniform
averaging is novel. Also non trivial work was needed to extend previous results to encompass both tail averaging and
mini-batching, specifically to control the variance of SGD. Regarding the experiments, we feel that they are on par
with those included in most theoretical papers on the same topic at venues like NeurIPS, COLT, ICML, or AISTATS.
Their purpose is to illustrate theoretical findings, rather than providing a thorough empirical analysis. Finally, let us
respectfully point out that the second set of experiments illustrated on Figure 1.b-c actually illustrates the interplay
between 3 parameters: tail-averaging, step size, and batch size. We opted to focus on the effects of tail-averaging since
we felt that this emphasizes the main novelty of our theoretical analysis.

**Response for Reviewer #5.**     Regarding the experiments, we feel that they are on par with those included in most
theoretical papers on the same topic at venues like NeurIPS, COLT, ICML, or AISTATS. Their purpose is to illustrate
theoretical findings, rather than providing a thorough empirical analysis. We expect the conclusions of the experimental
section to hold for large real datasets as well, since such data tends to verify our Assumption 2 for large $r$. Regarding
other convex and smooth losses such as the cross-entropy or the logistic loss, we expect most results to hold but proofs
will be considerably different and details might change.

[Meta-Review · NeurIPS 2019]

This paper analyses the convergence property of SGD on RKHS under the setting of multiple passes, mini-batching and tail averaging. It is shown that the tail averaging gives faster convergence than the uniform averaging. Moreover, the mini-batch technique enables more aggressive step size. The paper is well written so that the effect of each technique is concisely summarized. This paper adequately put the paper itself in the literature. Several audiences would be interested in this paper.